# AI-driven discovery of synergistic drug combinations against pancreatic cancer

Mohsen Pourmousa[1,5], Sankalp Jain[1,5], Elena Barnaeva[1], Wengong Jin [2], Joshua Hochuli[3], Zina Itkin[1], Travis Maxfield[3], Cleber Melo-Filho [3], Andrew Thieme[3], Kelli Wilson [1], Carleen Klumpp-Thomas [1], Sam Michael [1], Noel Southall [1], Tommi Jaakkola [2], Eugene N. Muratov [3,4], Regina Barzilay[2], Alexander Tropsha [3,4], Marc Ferrer [1] & Alexey V. Zakharov [1]✉

Pancreatic cancer treatment often relies on multi-drug regimens, but optimal combinations remain elusive. This study evaluates predictive approaches to identify synergistic drug combinations using a dataset from the National Center for Advancing Translational Sciences (NCATS). Screening 496 combinations of 32 anticancer compounds against the PANC-1 cells experimentally determined the degree of synergism and antagonism. Three research groups (NCATS, University of North Carolina, and Massachusetts Institute of Technology) leverage these data to apply machine learning (ML) approaches, predicting synergy across 1.6 million combinations. Of the 88 tested, 51 show synergy, with graph convolutional networks achieving the best hit rate and random forest the highest precision. Beyond highlighting the potential of ML, this work delivers 307 experimentally validated synergistic combinations, demonstrating its practical impact in treating pancreatic cancer.

Pancreatic cancer exhibits significant genomic heterogeneity, with studies identifying an average of 63 genetic aberrations across 12 functional pathways[1,2]. Pancreatic ductal adenocarcinoma (PDAC), the most prevalent form, poses a unique therapeutic challenge due to its exceptional resistance to systemic treatments compared to other solid tumors[3]. Prognosis of pancreatic cancer in the past ten years has remained unchanged[4], and treatments have become ineffective[5] owing to growing resistance. Thus, discovering anti-pancreatic cancer agents with high efficacy and low toxicity is crucial and constitutes one of the most challenging tasks for the scientific community working in oncology.

Drug combinations have become standard therapy for numerous diseases including malaria, HIV, tuberculosis, and cancers[6]. Combination therapies can offer benefits such as enhanced efficacy, reduced toxicity, and avoiding the acquisition of monotherapy resistance[7]. Effective drug combinations have been found in clinical settings[8], with

rising preclinical efforts to identify synergistic pairs[9]. Approved drugs, with known toxicity profiles and large-scale availability, serve as ideal candidates for these discoveries. Employing combination therapy in the context of pancreatic cancer addresses the disease's heterogeneity and adaptive nature, leading to enhanced treatment efficacy and offering a promising strategy to overcome resistance mechanisms[3]. Though a few computational models have been reported in the literature recently[10,11], the extent of combination synergy in the context of pancreatic cancer remains largely unexplored.

The process of identifying effective combinations remains challenging due to the sheer number of available and potential drug-like molecules[12], the quadratic number of possible combinations, and the time-intensive nature of experimental validation. While high-throughput screening (HTS) of combinations is an established method for this task[13], the vast number of unique chemical combinations still limits its efficiency. This challenge applies not

[1]National Center for Advancing Translational Sciences (NCATS), National Institutes of Health, 9800 Medical Center Drive, Rockville, MD 20850, USA. [2]Computer Science and Artificial Intelligence Laboratory, Massachusetts Institute of Technology, Cambridge, MA 02139, USA. [3]Laboratory for Molecular Modeling, Division of Chemical Biology and Medicinal Chemistry, UNC Eshelman School of Pharmacy, University of North Carolina, Chapel Hill, NC 27599, USA. [4]Predictive, LLC, Raleigh, NC 27614, USA. [5]These authors contributed equally: Mohsen Pourmousa, Sankalp Jain. ✉e-mail: alexey.zakharov@nih.gov

only to pancreatic cancer but to many diseases that require combination therapies. In this context, in silico methods present an efficient complement to high-throughput screening (HTS) and in vitro screening[14].

One of the most popular computational drug discovery approaches, namely, Quantitative Structure-Activity Relationship (QSAR) modeling, employs statistical or machine learning approaches to establish and validate correlations between computed molecular features and experimentally measured biological activities of molecules[15–17]. This approach has seen widespread application in cancer drug discovery[10], using methods that range from simple linear[18] to non-linear machine learning methods, such as Neural Network (NN)[19], Support Vector Machine (SVM)[20] or Random Forest (RF)[21]. Historically, QSAR efforts focused on monotherapies, but recent advancements have shifted towards predicting drug synergies. Cheng et al. [22]. introduced a QSAR-based biological network proximity measure to predict drug synergy in hypertension and cancer. Other studies have leveraged machine learning and deep learning techniques for synergy prediction[23,24], with Preuer et al. [25]. demonstrating the superiority of deep learning over traditional models using a large oncology screen[26]. The DREAM AstraZeneca-Sanger Drug Combination Prediction Challenge provided a comprehensive combinatorial cell line screening dataset, incorporating molecular data such as somatic mutations, copy-number alterations, DNA methylation, and gene expression profiles, as well as compound data including putative drug targets and chemical properties[27]. Jin et al. introduced ComboNet, a deep learning architecture that jointly models molecular structure and biological targets. This model predicted 30 combinations, and testing revealed two synergistic combinations in the context of COVID-19 (7% hit rate)[28]. Other studies have introduced computational methods like DrugComboRanker[29] DIGRE[30] and RACS[31] to effectively predict drug pairs for experimental validation. These methods rely on various data types like known disease pathway interactions, post-treatment gene expression, or drug–protein interactions, which can limit their generalizability and predictive accuracy[27].

This study combines efforts from three independent groups– National Center for Advancing Translational Sciences (NCATS), The University of North Carolina at Chapel Hill (UNC), and the Massachusetts Institute of Technology (MIT)–leveraging various ML methodologies to discover new drug combinations against pancreatic cancer. NCATS initiates the study by conducting cell-based assays, screening 1785 single-agent compounds and identifying the 32 most active ones. NCATS then screened 496 combinations, testing all-vs.-all combinations of the 32 compounds. The three teams use these 496 screening results to train ML models, independently predicting the top 30 synergistic combinations from a virtual library of 1.6 million combinations. Finally, NCATS tests the synergy of the predicted combinations in cell-based assays, which reveals an average hit rate of 60% across the teams. This study not only demonstrates the effectiveness of ML models in predicting synergy but also reports 307 synergistic combinations in PANC-1 cells linked to multiple (mechanisms of action) MoAs. By way of outline, the *Results* section presents all findings, starting with the *Data overview*, followed by *Modeling Results* from NCATS, UNC, and MIT. It then assesses the predicted combinations experimentally in *Experimental validation of the predictions* and examines the MoAs in *Exploring the biological relevance of the most synergistic combination against PANC-1*. The *Discussion* section addresses both computational and experimental aspects of the study.

## Results
### Data overview
Figure 1 presents an overview of the workflow in this study, combining computational and experimental approaches. Three distinct modeling approaches led to the nomination of 88 combinations.

The study began by selecting 32 compounds from a library of 1,785. Figure 2a illustrates the activity distribution of these compounds in single-agent dose–response curves in PANC-1 cell assays. The $IC_{50}$ values range between 2 nM and 3 μM, demonstrating variability in activity values. Screening all possible pairwise combinations of 32 compounds, with titration producing 10×10 matrices, generated a modeling dataset of 496 combinations. Conducting the screenings in duplicates enabled the analysis of assay result reproducibility across several synergy metrics (Supplementary Table 1). Figure 2b and Supplementary Fig. 1 present the correlation of the gamma, beta, and Excess HSA scores. While all metrics confirmed the reproducibility of the assay results and matrix screening technology, the higher correlation of gamma scores led to its selection as the synergy metric. Machine learning used the average gamma scores of each combination for model training. In addition to gamma, the training data includes SMILES representations, IC50 values, and MoAs for individual compounds.

This study focuses on identifying synergistic combinations, disregarding the distinction between additivity and antagonism. Each group developed models using their own methodologies, whether by using gamma scores from the training data or creating bespoke synergy metrics from IC50 values. Nevertheless, the study defines a precise cutoff to evaluate model performance in prospective experiments: Gamma scores below 0.95 indicate synergism, while scores above 0.95 signify non-synergism[32].

The original library of 1785 compounds produces 1,592,220 combinations. Excluding the 496 combinations from the training set reduces the total to 1,591,724 combinations for the test set. Although the test set omits specific combinations from the training set, it still includes combinations involving compounds from that set. The test dataset includes SMILES representations, IC50 values, and MoAs, but does not include gamma values. Modeling focused on predicting synergy for the test combinations before experimental validation. Three independent research groups conducted separate modeling approaches, each delivering three lists of top 30 synergistic combinations. The selected combinations underwent experimental testing to evaluate the models' accuracy.

### NCATS modeling results
Supplementary Table 2 provides detailed statistics about the models. The cross-validation scheme followed the one-compound-out approach, resulting in 32 splits. None of the machine learning methods (RF, XGBoost, and DNN) outperformed the others when using comparable descriptor types. On the other hand, RF and XGBoost models using only Avalon or Morgan fingerprints achieved significantly higher area under the curve (AUC) values for receiver operating characteristic (ROC) plots compared to models using only RDKit or in-house descriptors. Most models combined various descriptor types, with AUC values nearing 75%. However, the combining approach did not improve performance compared to using single fingerprints alone. The top-performing model used Avalon-2048 fingerprints and combined RF classification with regression, achieving an AUC of $0.78 \pm 0.09$ (Fig. 3). The model predicted and selected the final top 30 compounds with the highest synergy probabilities.

### UNC modeling results
Supplementary Table 3 provides detailed statistics about the models. Two cross-validation approaches evaluated models' performances, one-compound-out[33,34] and everything-out[35] validation. The one-compound-out strategy, which evaluates combinations with one training compound and one non-training compound, resulted in most models achieving a correct classification rates (CCRs) between 0.6 and 0.65, with a positive predictive value (PPV) between 0.6 and 0.7. The everything-out strategy, which evaluates combinations with no training compounds, produced CCRs between 0.5 and 0.55, and PPVs

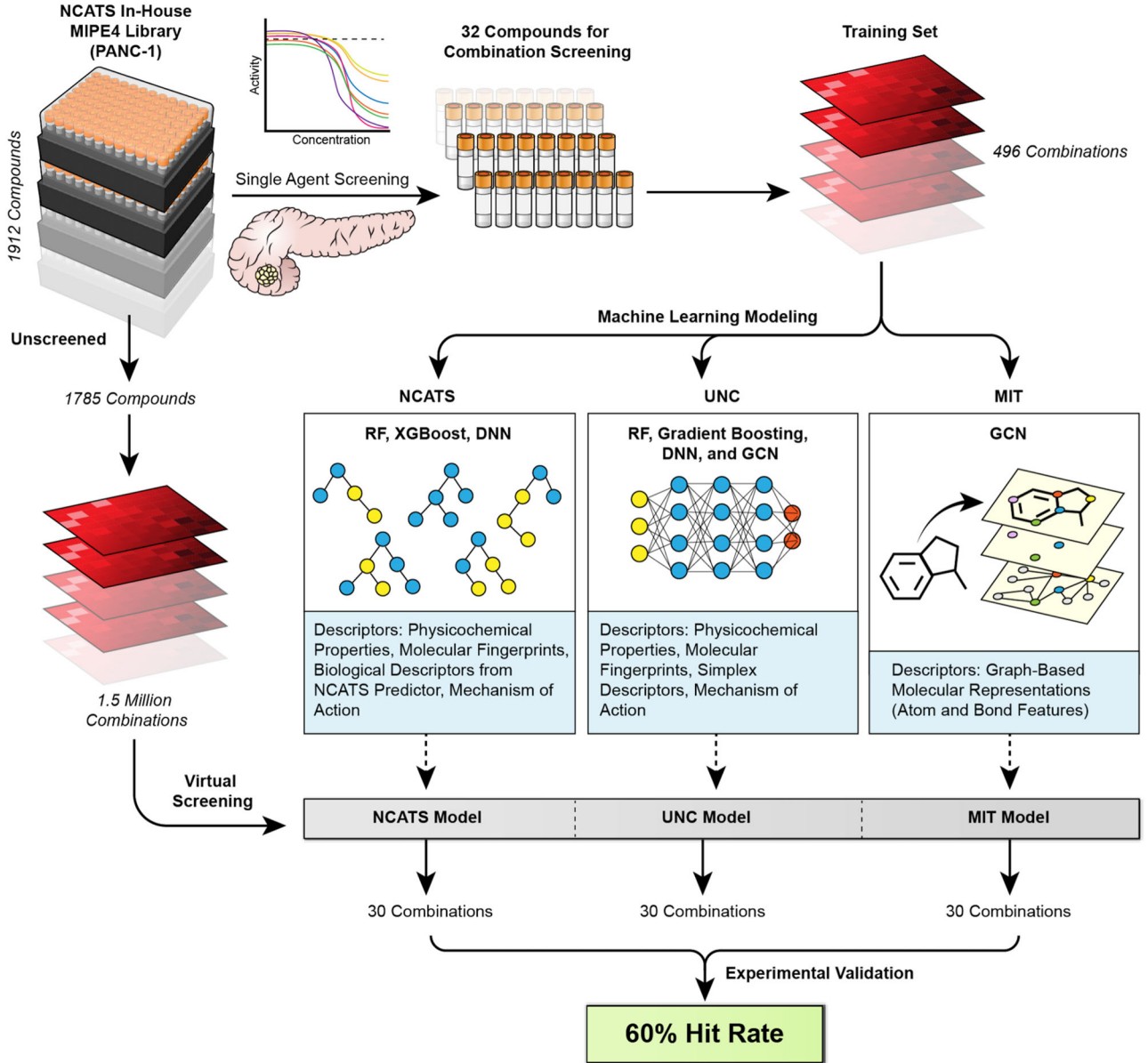

**Fig. 1 | Workflow illustrating the combination of computation and experiment to discover synergistic drug combinations.** High-throughput screening of 1785 single-agent compounds from the MIPE4 library in PANC-1 cells identified 32 active compounds. All-vs.-all combinations of these 32 compounds generated synergy data for 496 combinations. The synergy data served as a training dataset for three machine learning teams. NCATS used Random Forest (RF), XGBoost, and Deep Neural Network (DNN); UNC used RF, gradient boosting, DNN and graph convolutional networks (GCN); MIT used GCN. Training features included molecular descriptors, such as Avalon and Morgan fingerprints, RDKit descriptors, and biological features. Models predicted synergies across 1.6 million virtual drug combinations, and the top 30 combinations from each team were experimentally tested in PANC-1 cells, achieving an average 60% hit rate.

between 0.5 and 0.6. This difference reflects the added challenge of the everything-out strategy. Y-randomization for both strategies produced validation statistics consistent with random predictions.

Models incorporating experimental $IC_{50}$ values in their chemical representation did not significantly outperform those that omitted them. Furthermore, descriptor composition strategies, whether averaging or summing columns, performed similarly among the descriptor-based models.

UNC developed three models using only descriptors and two models combining descriptors with IC50 data. Consensus model predictions were calculated as an average between the three descriptor consensus model predictions, and the two descriptors + IC50 individual model predictions. The top 30 nominated combinations did not come directly from the top 30 ranked lists of either the consensus model or the descriptor-only model. Rather, in addition to model

predictions, the selection process followed three supplementary criteria. Table 1 lists these three mutually-exclusive selection tiers. The IC50 values of compounds from the test set served as one criterion. Another required that one compound in the pair exists in the training set. The third criterion ensured that the combination corresponds to the most synergistic MoA pairs from the training set.

Using these criteria, the first tier included combinations with consensus scores above 0.7, two active compounds, one compound in the training set, and synergistic MoA pairs, resulting in 12 combinations. The second tier selected the top combinations ranked by descriptor-only predictions, with two active compounds, one compound in the training set, and no synergistic MoAs, adding another 12 combinations. The third tier identified the top six combinations ranked by the consensus model, featuring two active compounds, no compounds from the training set, and synergistic MoA pairs.

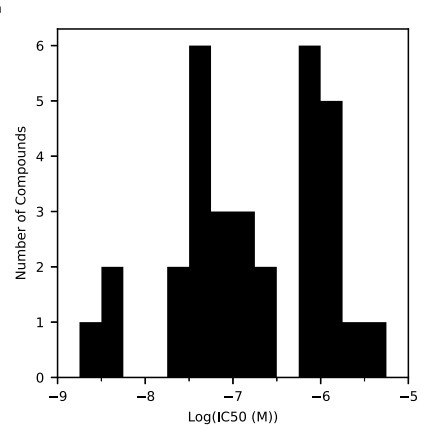

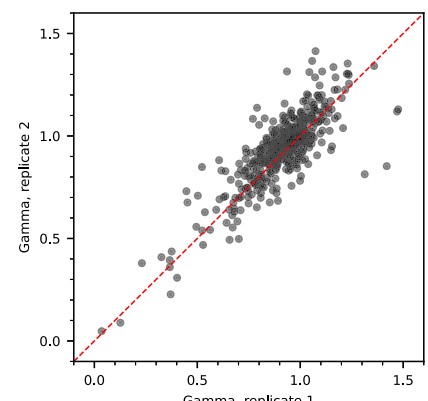

**Fig. 2 | Analysis of the variation and reproducibility of PANC-1 training dataset.** **a** Variation of activities of 32 compounds in single-agent dose–response curves. Log(IC$_{50}$) varies between −8.7 and −5.5 ($\approx 2$ nM–3 μM). **b** Reproducibility of experiments. Gamma values of 496 combinations in two replicates have a Pearson's coefficient of 0.83. Combinations with Gamma <0.95 are considered synergistic. Similar plots for other synergy metrics are in Supplementary Fig. 1. Source data are provided as a Source Data file.

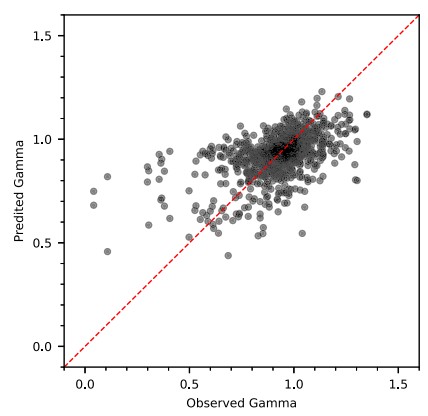

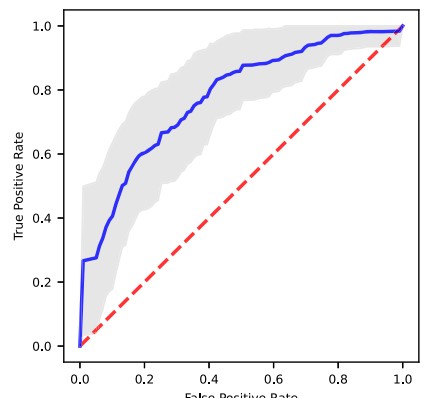

**Fig. 3 | NCATS' random forest models to predict Gamma.** Training set included 496 combinations constituted from 32 drugs (all vs. all). Cross validation strategy was one-compound-out, hence 32 folds. Average of Avalon 2048 fingerprints of each pair featurized the combination as a single 2048-dimensional vector. **a** Regression. The plot overlays predictions for 32 validation sets (withheld within each of 32 folds) with a Pearson's coefficient of 0.52. **b** Classification. Model yields 32 ROC plots (not shown for clarity) with average AUC and standard deviation of 0.78 ± 0.09. Blue line, average ROC plot; grey area, ±1 standard deviation; red dashed line, random prediction baseline; Gamma < 0.95, synergistic; Gamma ≥ 0.95, non-synergistic. Similar plots for Morgan 2048 and RDKit descriptors are in Supplementary Figs. 2 and 3. Source data are provided as a Source Data file.

## MIT modeling results

Supplementary Fig. 4 shows the ROC plot for the graph convolutional model, which achieved an average test AUC of 0.840 ± 0.036, averaged across five different random splits. Supplementary Table 4 lists the AUCs for each split. Supplementary Fig. 5 displays the distribution of predicted synergy scores, and Supplementary Fig. 6 highlights the most frequent compounds and their average synergy scores. The model predicted synergy scores for 1,591,724 combinations and selected the top 30 while ensuring diversity by limiting the final selection to the top five combinations per compound. No compound appeared in more than five combinations.

## Experimental validation of predictions

Each institute independently proposed a top 30 list of combinations for testing. Figure 4 illustrates the compound combinations selected by each institute. The lists had minimal overlap, highlighting the independence of their development processes. Only two combinations appeared in both MIT's and UNC's top 30 selections. Experimental testing of the 88 unique combinations involved duplicated measurements in the PANC-1 cell line assay, using titrations to generate 10 × 10 dose–response matrices, similar to developing data for ML training.

Using the same gamma cutoff to define synergism (γ < 0.95), all models from the three institutes delivered remarkably high hit rates. The graph convolutional model from MIT excelled, correctly predicting 25 out of 30 compound combinations and achieving an 83% hit rate. The RF model from NCATS, which combined classification probabilities and regression pseudo-probabilities, followed with a 53% hit rate, correctly identifying 16 out of 30 combinations. UNC's models predicted 12 out of 30 combinations accurately, resulting in a 40% hit rate.

**Table 1 | UNC nomination strategy. Each successive tier nominates combinations not previously nominated by an earlier tier, ensuring the tiers remain mutually exclusive**

| Ranks | Count | Combination filters | MoA filter | Ranking models | Ranking criteria |
|-------|-------|--------------------|-----------|----------------|------------------|
| 1–12 | 12 | 2 active, 1 train | Yes | Consensus | score > 0.7 |
| 13–24 | 12 | 2 active, 1 train | No | Descriptor-only | 12 highest |
| 25–30 | 6 | 2 active, 0 train | Yes | Consensus | 6 highest |

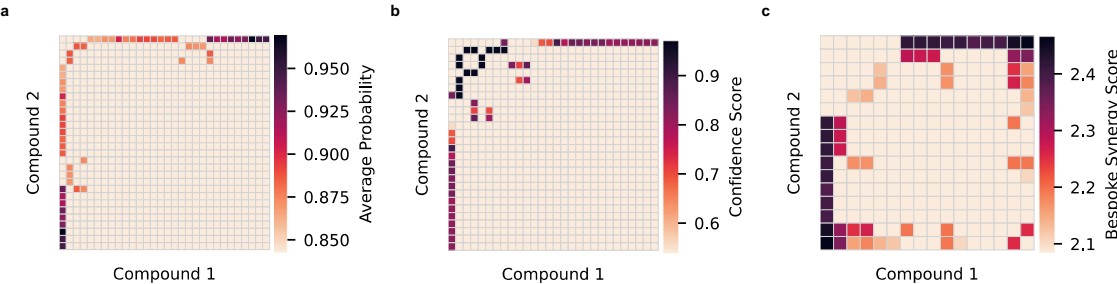

**Fig. 4 | Predicted synergy of unseen combinations by three independent models. a** NCATS, (**b**) UNC, (**c**) MIT. Each institute nominated 30 combinations, resulting in 60 synergy values per a symmetric matrix. Voxels represent scores between compounds on Compound 1 and Compound 2 axes. Color bars represent ranges of different synergy scores adopted by each team. Compound names, excluded for clarity, are accessible in the source data, provided as a Source Data file.

**Table 2 | Performance of three models and a consensus model in predicting synergy of 88 nominated combinations**

|  | TP | TN | FP | FN | Sens | Spec | PPV | NPV | Balanced Accuracy | AUC |
|---|---|---|---|---|---|---|---|---|---|---|
| NCATS | 40 | 15 | 22 | 11 | 0.78 | 0.41 | 0.65 | 0.58 | **0.59** | 0.56 |
| UNC | 50 | 0 | 37 | 1 | 0.98 | 0.0 | 0.57 | 0.00 | 0.49 | 0.60 |
| MIT | 51 | 2 | 35 | 0 | 1.0 | 0.05 | 0.59 | 1.00 | 0.53 | **0.78** |
| Consensus | 51 | 2 | 35 | 0 | 1.0 | 0.05 | 0.59 | 1.00 | 0.53 | 0.67 |

This analysis follows experimental testing of 88 combinations, with each group reporting their prediction labels for all combinations, including the 30 they initially provided. Values in bold indicate the models with the highest Balanced Accuracy (NCATS) and AUC (MIT) among others. *TP* True Positives, *TN* True Negatives, *FP* False Positives, *FN* False Negatives, *Sens* Sensitivity, *Spec* Specificity, *PPV* Positive Predictive Value, *NPV* Negative Predictive Value.

With experimentally-derived synergy values for 88 combinations, a retrospective evaluation assessed how well the final models from each institute predicted these combinations. In other words, each group reported their prediction labels for all 88 combinations, including the 30 they initially provided. Table 2 and Supplementary Fig. 7 present the results. The hit rates shifted as the models now predicted 88 combinations instead of the original 30, with MIT, UNC, and NCATS achieving 58%, 57%, and 45%, respectively. The NCATS model achieved the highest balanced accuracy (BACC = 0.59) and produced the fewest false positives (FP), resulting in the highest precision at 0.65. Although MIT provided the highest AUC value (AUC = 0.78), it generated a high number of FPs. Finally, the consensus of all three approaches did not significantly improve precision.

### Exploring the biological relevance of the most synergistic combination in PANC-1

With the experimental synergy values available, this section shifts its focus from modeling to analyzing the experimental results. Out of 496 training and 88 test combinations, 256 and 51, respectively, have experimentally-determined gamma scores below 0.95 (Fig. 5). These 307 synergistic combinations include several recurring compounds and MoAs. As shown in Fig. 6, Carfilzomib is the most prevalent compound. Notably, combinations involving Carfilzomib exhibit the highest synergy. A similar analysis of MoAs reveals that proteasome inhibition is the most common, followed by Polo-like kinase 1 (PLK1) and histone deacetylase (HDAC) inhibition. HDAC inhibition, despite being less frequent than PLK1 inhibition, results in more favorable Gamma scores.

Previous analyses considered all synergistic combinations with Gamma scores below 0.95 (distribution shown in Fig. 5). Among the 307 combinations, 26 exhibit strong synergy with Gamma scores below 0.5. These span 20 different MoAs, as detailed in Supplementary Table 5. Network analysis (Fig. 7) identifies proteasome-HDAC inhibition as the most frequent synergistic combination, a trend consistent even when considering all synergistic combinations with Gamma scores below 0.95 (Supplementary Fig. 8). Other significant synergistic MoAs in the set of 26 combinations, excluding proteasome inhibition,

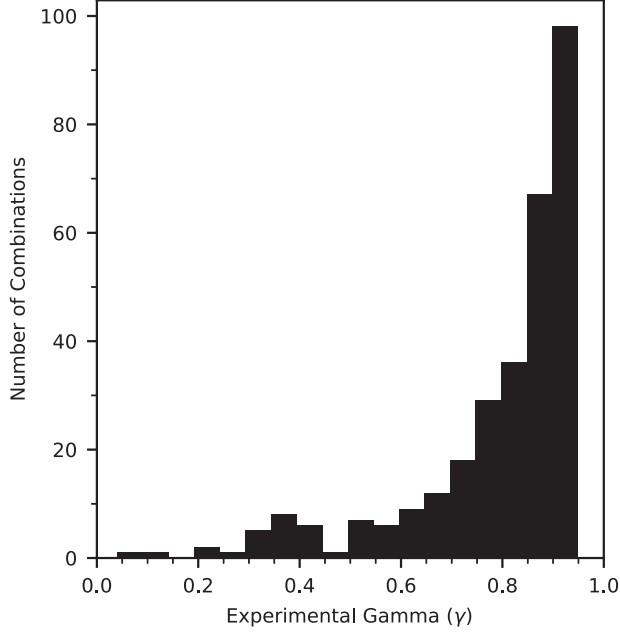

**Fig. 5 | Distribution of Gamma for 307 experimentally-validated synergistic combinations.** Most scores are close to 0.95. 26 combinations exhibit strong synergy with scores below 0.5. Source data are provided as a Source Data file.

include HDAC inhibition combined with mutant-p53 activation and HDAC–Survivin inhibition.

Assessing the statistical significance of the emerging MoA–MoA interactions in synergistic combinations remains important. Proteasome–HDAC inhibition occurred 9 times within the 307 synergistic combinations. If this interaction proves the most prevalent, it encourages further exploration to generalize the findings and predict other compounds with these two MoAs. Alternatively, the observed MoA–MoA interactions may have emerged by chance. To test the significance, the process involved comparing these

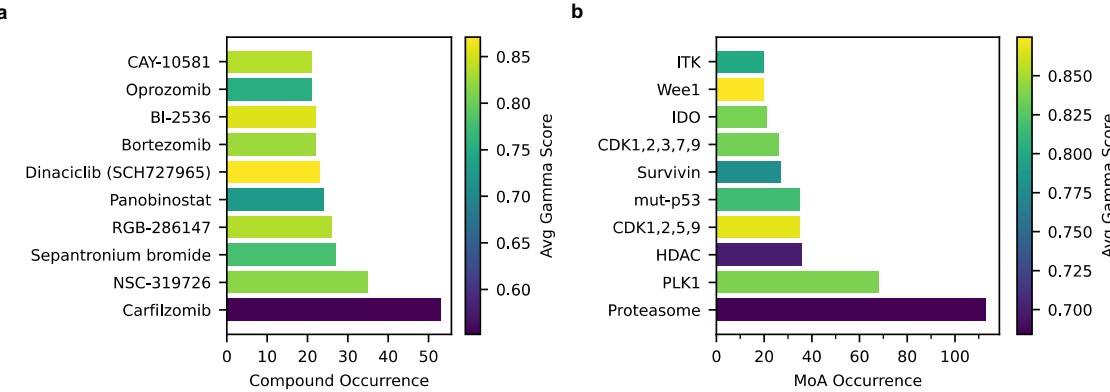

**Fig. 6 | Number of occurrence of compounds and MoAs among 307 experimentally-validated synergistic combinations colored by average synergy. a** Compounds, (**b**) MoA. Each MoA in (**b**) is abbreviated for simplicity; for example, "HDAC" refers to "HDAC Inhibitor", where HDAC is a protein name. Supplementary Fig. 8 provides details of abbreviations. Source data are provided as a Source Data file.

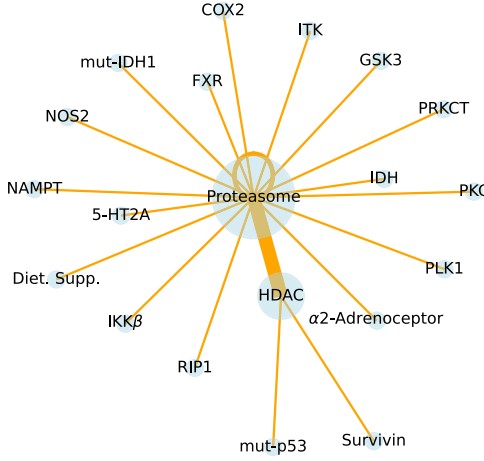

**Fig. 7 | Network analysis of MoAs in 26 strongly synergistic combinations (gamma < 0.5).** Node sizes and edge widths are proportional to MoA and MoA–MoA frequency, respectively. Each MoA is abbreviated for simplicity; for example, "HDAC" refers to "HDAC Inhibitor", where HDAC is a protein name. Supplementary Fig. 8 provides details of abbreviations. Source data are provided as a Source Data file.

results against random combinations. The analysis took the 64 unique compounds from the 307 synergistic combinations and generated 307 random combinations, repeating this 1000 times. Proteasome–HDAC inhibition occurred 4.4 times on average, with a standard error of 1.9 (see Supplementary Fig. 9 for a random network map). This comparison shows that the 9 occurrences in the synergistic compounds significantly exceed random expectations ($P$-value < 0.01). Proteasome–HDAC inhibition gains even more importance because it remains the most frequent interaction with a stricter synergy criterion ($\gamma$ < 0.5; see Fig. 7).

## Discussion

The complexity of pancreatic cancer drove this study to uncover synergistic drug combinations through an integrated approach of computational and experimental methods. Three independent institutes developed machine learning models and identified 30 drug combinations for PANC-1 cells, with only two overlapping selections, underscoring the complementary nature of their distinct methodologies. This collaboration achieved notable hit rates of 83%, 53%, and 40%, utilizing graph convolutional networks (GCN), a combination of RF classification and regression, and a customized selection process with a range of algorithms and features. The study reports 307

experimentally validated synergistic combinations. Among these, proteasome inhibition, especially when combined with HDAC inhibition, emerged as a recurring and highly synergistic MoA. Since the study integrates computational approaches with experimental validation, several discussion points arise for each category.

Both MIT and UNC used graph convolutional networks, but MIT's superior model performance likely resulted from training on auxiliary data from NCI-ALMANAC, which includes drug synergy data from multiple diseases (4000 data points). Moreover, UNC incorporated graph convolutional networks only as part of a broader modeling approach and selection process. The MIT model also stood out for its diverse selection of synergistic combinations. Figure 4 illustrates that the NCATS and UNC models frequently identified the same compounds as synergistic with multiple others. To ensure diversity, MIT deliberately limited the final selection to the top five combinations for each compound, leading to a more varied top 30 list with fewer recurring compounds.

The nomination process, in addition to the algorithms and features, influences the performance of the models. For example, NCATS likely achieved a high PPV by not excluding combinations with recurrent compounds, selecting combinations where a single recurrent compound primarily drives the synergy. In contrast, MIT limited its selection to the top five combinations involving each compound. UNC's three-tier selection process may have contributed to its lower specificity and balanced accuracy.

Comparing the ML results from this study with existing literature provides valuable insights. NCATS observed similar performance among RF, XGBoost, and DNN (see Supplementary Table 2), with a noticeable decline when compound features shifted from Avalon or Morgan fingerprints to RDKit or in-house descriptors. This highlights that, among these three algorithms, feature selection holds greater significance than the algorithm itself. Celebi et al. [27]. also reported that XGBoost and RF outperformed linear regression, Lasso, and SVM. Therefore, both studies demonstrate that XGBoost and RF are suitable algorithms for predicting synergy. However, this result contrasts with a study that trained a DNN on a large oncology screen, demonstrating the advantage of deep learning over standard ML models like RFs and SVMs[25]. In general, making direct comparisons between this study and others is challenging due to differences in feature sets and endpoints. For example, the preceding study used chemical and genomic information as input, Celebi et al. [27]. used data from the DREAM AstraZeneca-Sanger Drug Combination Prediction Challenge, which incorporates a wider range of information from different cell lines, while the current study used fingerprints from SMILES, MoAs, and single-agent activities in PANC-1 cells. In a similar context, the methods presented in this study may not perform

equivalently across different datasets, such as those from other disease areas. Despite the challenge of directly comparing with previous models, this work demonstrates a key strength by rigorously assessing the models' generalizability across 1.6 million combinations, resulting in remarkable hit rates.

Among prospectively discovered combinations, NSC-319726 and AZD-8055 showed synergies with several compounds, highlighting their potential clinical relevance. NSC-319726–AZD-8055 combination has an experimentally-determined gamma score of 0.78 (Supplementary Fig. 10). NSC-319726, a preclinical activator of mutant p53, is under investigation for its potential to treat pancreatic cancer[36]. Mutant p53 is a critical driver in pancreatic tumor development, making it a key target for therapeutic intervention. Significant progress has been made in the development of p53 activators, particularly MDM2 antagonists, which work by stabilizing and reactivating p53's tumor-suppressing function[37,38]. Notably, MDM2 antagonists like idasanutlin have reached phase III clinical trials, highlighting their potential as promising treatment options for pancreatic cancer[39,40]. Similarly, AZD-8055, an mTOR inhibitor, is under investigation for its potential role in treating various cancers, including pancreatic tumors. mTOR inhibitors like AZD-8055 target a critical pathway involved in cancer cell growth and survival[41]. However, one of the effects of mTOR inhibition is the activation of autophagy, a process that cancer cells can use to survive under stress. To counteract this, autophagy inhibitors, such as chloroquine and hydroxychloroquine, have been explored in combination with mTOR inhibitors in clinical trials for pancreatic cancer. Unfortunately, the results have been less encouraging, as clinically safe doses of autophagy inhibitors have not shown significant benefits in the advanced stages of the disease[42,43]. This may be due to the inability to achieve sufficient autophagy inhibition at doses that are safe for patients[44].

Testing for general cytotoxicity is crucial for assessing therapeutic compounds and combinations. The synergistic combinations in this study demonstrate efficacy in PANC-1 cell lines, though it does not include further biological validation. Future research could explore cytotoxicity by using control cell lines and more advanced models. Organoids from pancreatic cancer patients and normal tissue could provide a more accurate representation of the tumor microenvironment, refining evaluations of drug efficacy and specificity. Spheroid cultures, which mimic the three-dimensional structure of tumors, could also help assess compound penetration and activity. In vivo studies with pancreatic cancer mouse models could further evaluate therapeutic potential, observing effects on tumor growth, metastasis, and potential toxicity. These advanced models would guide future research toward clinical trials by providing comprehensive safety and efficacy assessments.

The findings of this collaborative research offer several key implications. First, they highlight the effectiveness of machine learning models in predicting drug synergy. Second, they underscore the value of prospective validation, even for resource-intensive matrix screening, providing a reliable assessment of models' predictive power in real-world scenarios. Third, this research serves as a compelling case study for benchmarking common machine learning approaches of varying complexity in a carefully designed prospective investigation. Fourth, it demonstrates the power of collaborative research, where distinct machine learning methods complement each other's hit lists. Finally, advancing the combinations or MoAs uncovered here in preclinical studies has strong potential to drive the development of effective therapies for pancreatic cancer.

## Methods

This research complies with all relevant ethical regulations. As the study involved only commercially available cell lines (PANC-1, ATCC catalog number CRL-1469), no additional ethical approval was required.

### Generation of the training dataset

We used NCATS' in-house compound collection MIPE4 which consists of nearly 2000 antineoplastic compounds with diverse and redundant mechanisms of action (MoA), and includes approved drugs, phases I – III investigational drugs, and pre-clinical molecules. The screening of this library and all following in vitro experiments were performed with continuous tumor-cell line from a human carcinoma of the exocrine pancreas (PANC-1, ATCC catalog number CRL-1469) that is often used as a model for this highly aggressive type of pancreatic cancer.

The initial screening of MIPE4 library was performed in a duplicated single-agent dose-response manner (11 serial dilutions 1:3 in the range of [46 uM – 0.78 nM]. For this assay, PANC-1 cells were grown in DMEM media with 10% FBS and 1% antibiotics mix. The day of the experiment, they were harvested with 0.25% Trypsin for 5 min, spun down and seeded as a cell suspension in growth media 500 cells/5ul/well. The cell density was experimentally selected based on efficient proliferation rate of cells for the period of 72 h. Cells were plated onto white tissue culture-treated 1536-well Aurora plates (Brooks Automation, Chelmsford, MA) with MultiDrop Combi dispenser (Thermo Scientific, Logan, UT), and allowed to attach overnight at 37 °C, 5% CO$_2$. Next, 23 nl/well of compound solutions in DMSO were added to cells with a pintool transfer (Kalypsis, San Diego, CA). As the positive control, we used "no cells" column which would be equal to 100% killing capacity of a compound (IC$_{100}$), and DMSO was used as a vehicle neutral control (IC$_0$). The cells were stimulated with compounds or DMSO for 72 h at 37 °C, 5% CO$_2$, 95% humidity. For detection, we used CellTiter-Glo® Luminescent Cell Viability Assay (Promega, G7573), an ATP-based detection method of viable cells. This assay applies the properties of a thermostable firefly luciferase to generate a stable "glow-type" luminescent signal which could be measured within minutes when added directly to cell culture.

At the end of 72 h stimulation period, 5 ul/well of CellTiter-Glo® reagent was added to plates with BioRAPTR FRD dispenser. The plates were incubated at ambient temperature for 10 min and then the luminescent signal was read on ViewLux plate reader (PerkinElmer, Waltham, MA) with 3 s exposure time.

Screening the MIPE library in proliferating PANC-1 cells revealed 32 compounds. These compounds, with IC50 values in Fig. 2, show variability and potency, reach efficacy values of at least 50%, and follow curve classes 1.1 and 1.2[45]. The 32 compounds produce $32 \times 31/2 = 496$ combinations, each tested in $10 \times 10$ matrices with nine 1:2 serial dilutions. The concentration range for each compound was selected individually to arrange their IC$_{50}$ roughly in the middle of that range. Each 1536 well plate included a DMSO control (IC$_0$) and Bortezomib, a well-known cancer drug, as the positive control (IC$_{100}$). An example of a screening plate, including the response to negative and positive controls, is provided in Supplementary Fig. 11. Each block was tested in duplicate. Both the generation of training data and the validation of ML predictions used the PANC-1 cell line $10 \times 10$ assay procedure in duplicates.

To perform the matrix screening, the compounds mixtures were pre-dispensed to empty 1536-well white solid bottom plates with Echo 650 acoustic liquid handler (Beckman Coulter), 20 nl/well each. Then, the plates were dispensed with PANC-1 cell suspension at 500 cells/5ul media density and placed in a tissue-culture incubator (5%CO$_2$, 37 °C, 95% humidity). CellTiter-Glo® reagent was added to wells 72 h later for detection as described above.

### Compounds standardization

Chemical structures of molecules in the training and test mixtures were standardized and curated following our canonical data curation practices using ChemAxon Standardizer version 20.9[46]. Briefly, salts and solvents were stripped from all compounds followed by removal of counterions, large organic compounds (Da ≥ 2000), mixtures, and

inorganic compounds. Specific chemotypes such as aromatic, nitro groups, sulfo groups, tautomers, and protonation states were standardized using ChemAxon Standardizer software (https://chemaxon.com/). As chemical standardization is the first step in obtaining a curated dataset for modeling, we used ChemAxon Standardizer for structural analysis in both descriptor-based and graph-based approaches.

## Synergy quantification

The degree of combination synergy is quantified by comparing the observed compound combination response against the expected response, calculated using a reference model that assumes no interaction between compounds. The commonly-used reference models include the highest single agent (HSA)[47], Bliss[48], Loewe[49] and Zero interaction potency (ZIP) model[50]. This study analyzed different synergy metrics, such as gamma (γ), beta, and Excess HSA, to assess the reproducibility of duplicated combinations. It chose gamma for its lower sensitivity to variability in duplicates. A previous study determined $\gamma < 0.95$ for synergism through isobolographic assessment of three ibrutinib-drug combinations[32]. Consistently, this study classifies combinations with Gamma scores below 0.95 as synergistic for the ML models, while higher scores indicate non-synergism. The models predicted synergy without differentiating whether non-synergistic combinations were independent or antagonistic.

## Computational modeling approach 1: NCATS

For machine learning, the molecular structures can be represented as numerical feature vectors, or molecular descriptors[51]. We employed several features and their concatenations to benchmark different machine learning methods. Each combination had two compounds and therefore two feature vectors were averaged to yield a single feature vector per combination. Avalon1024, Avalon2048, Morgan1024, and Morgan2048 fingerprints and RDKit descriptors (https://rdkit.org) were generated using python RDKit package. Additionally, an in-house Python 3.7.7 was used to generate biological descriptors from NCATS predictor (https://predictor.ncats.io/)[52]. Mechanisms of action were turned into feature vectors as follows: There were 821 unique mechanisms of action for the original 1785 PANC1 screening compounds. Each mechanism of action was encoded as an element in a vector of length 821, and the presence or absence of a mechanism of action for each combination was represented by 1 and 0, respectively. For instance, a combination with two distinct mechanisms of action would be encoded as a vector that has two elements set to 1 and the rest of them set to 0.

Random forest (RF) classification and regression[53], eXtreme Gradient Boosting (XGBoost)[54], and deep neural network (DNN)[55] were used for benchmarking machine learning algorithms. AUC of the ROC plot measured the performance of the classification models. XGBoost learning rates were tuned to be 0.01. DNN consisted of a sequential model with three hidden layers[56]. The main parameters were optimized sequentially as follows: learning rate=0.0001, optimizer=Adam, batch size=128, epochs=70. A nested hyperparameter tuning scheme optimized the hidden layers, testing 700, 1000, and 2000 neurons in the first layer, 500 and 700 in the second, and 200 and 300 in the third. The final DNN architecture used 700, 500, and 300 neurons in the respective layers.

Cross-validation scheme for training the model was one-compound-out[33,34]. This approach involves excluding one compound from each fold so that the fold has all the combinations except those containing that compound. As 32 compounds constituted the combinations (or training data), a total of 32 folds were performed within this scheme. The cross validation of the nested hyperparameter tuning in DNN split each 32 folds into three smaller folds.

Consensus modeling in general outperformed individual QSAR models[56]. NCATS developed consensus models using two approaches:

(i) averaged the probabilities of individual models and (ii) applied the majority vote from individual models. The final model used the first approach, converting regression values to pseudo-probabilities by scaling them to a range of 0 to 1 and then subtracting the scaled value from 1. This subtraction ensured that lower gamma values reflected stronger synergy or higher probability of being synergistic. The model then averages regression pseudo-probabilities and classification probabilities.

## Computational modeling approach 2: UNC

**General.** RDKit descriptors and Morgan Fingerprints for both training set compounds and virtual screening library were calculated using the RDKit package in Python. Simplex descriptors[57] were calculated with the HiT QSAR program[38]. Machine learning models were developed using the Python packages scikit-learn and PyTorch (for neural networks). Scikit-learn models used the default hyperparameters. All models were trained as classifiers (using binary labels) but produced a "confidence score" by different means. These "confidence scores" were averaged during consensus prediction calculation. Since the Simplex descriptors are computationally expensive[58], to calculate consensus predictions on the full test set, only the descriptor-based models with RDKit descriptors and Morgan Fingerprint were run to calculate consensus predictions on the full test set. Subsequently, the top 2000 scoring combinations were evaluated with the Simplex-based models and the models with $IC_{50}$ data.

## Mechanism of action selections

The training set columns 'MOA 1' & 'MOA 2' are aggregated and sorted, to form one MOA pair per row (one row per 496 unique drug pairs in the training set). Matrix rows are then grouped by unique MOA pairs. Counts of the number of rows from the original training set included, and the means for the 'label' (activity binary) column, were calculated for each of the 280 unique MOA pairs present in the training set. MOA pairs with 3 or more representative drug pairs in the training set, and a mean 'label' value greater than or equal to 0.66, were classified as the 24 most synergistic MOA pairs occurring in the training set. Chemotext[59], ROBOKOP[60] and knowledge graph mining[61] were used to identify and refine the MOAs of test compounds. Virtual screening of the test set (1.6 M unique drug pairs), to remove drug pairs which do not represent any of the above 24 MOA pairs, yielded a subset of 888 unique drug pairs prioritized for selection by MOA model.

## Pure descriptor models

A total of 15 model types trained purely on gamma binary labels were developed for use in consensus. Two descriptor types (RDKit descriptors and Morgan Fingerprint) were calculated for individual compounds in training mixtures and composited by either element-wise average or sum at every position in the descriptor. A third descriptor type, Simplex[62], was calculated for the training mixtures, and did not require composition. Each of these descriptor/composition pairs was used as input for neural network, random forest, and gradient boosting models[63]. To calculate a consensus, the prediction of every model for every descriptor was averaged. This resulted in three consensus models, each corresponding to a different descriptor.

## Descriptor and $IC_{50}$ models

Two models were developed incorporating compound $IC_{50}$ data: a graph convolutional model and a neural network model based on Simplex descriptors. To develop the graph model, standardized SMILES strings for each of the constituent compounds were converted into graphs (adjacency matrices and node feature matrices) using the OpenChem software package[64]. A combined mixture graph was created by taking the direct sum of individual adjacency matrices and the concatenation of node feature matrices. This combined graph was

passed through a graph convolutional network resulting in a latent representation of the mixture graph, to which the respective Log IC$_{50}$ values of each component compound were concatenated. The resulting representation was passed further through a neural network, and a classification score was produced as the final output.

For the neural network model, mixture Simplex descriptors were first produced and then cleaned, removing the highly correlated simplexes. The resulting Simplex descriptors were passed through several layers of a neural network, after which the Log IC$_{50}$ values of each component compound were appended to the mixture representation. Like the graph model, the resulting representation was used as an input to the neural network model, and a classification score was produced as output. Each model was defined as above up to a set of hyperparameters (number of layers, size of each layer, learning rate, number of training epochs, etc.). To find optimal hyperparameters for each model, a specific mixture-oriented external validation was performed on several models. For a wide range of hyperparameter values, only insignificant differences were observed between models. Therefore, within this range, the final models were chosen for computational efficiency.

### Computational modeling approach 3: MIT

MIT's ComboNet[28] approach utilizes a graph convolutional network that embeds a molecule into a continuous feature vector. This vector is learned automatically through the single-agent and combination data rather than hand-crafted. When trained on the single-agent data, the model takes a molecule as input and predicts whether it will inhibit cancer cell growth (binary label). When trained on the combination data, the model first takes a combination (A, B) as input and predicts the inhibition score c(AB) of this combination. Then it predicts their single-agent activity s(A) and s(B) and computes its expected inhibition score under bliss independence assumption: s(AB) = s(A) + s(B) - s(A)s(B). Finally, it quantifies the score of combination (A, B) as c(AB) - s(AB), herein referred to as bespoke synergy score, with higher values indicating stronger synergy. Ref. 28 provides additional details. The score assigns higher values to more synergistic combinations, unlike gamma. In addition to pancreatic cancer (PANC-1), our model is trained on 40 different cell lines collected from NCI-60 (single-agent data)[65] and NCI ALMANAC (combination data)[13]. Following a multi-task training scheme, our model is trained on 47 K single-agent and 4 K combination data points in total, which is much larger than the PANC-1 data. To evaluate our approach, we split the pancreatic cancer combination dataset into training (80%), validation (10%) and test sets (10%).

Importantly, in this adaptation of ComboNet, we omitted drug–target interaction data, a deviation from the initial design[28], as it did not enhance the model's predictive accuracy.

### Reporting summary

Further information on research design is available in the Nature Portfolio Reporting Summary linked to this article.

## Data availability

The dataset used for modeling is available in our GitHub repository (https://github.com/ncats/PANC1)[66]. Source data are provided with this paper.

## Code availability

The scripts used for the data analysis and modeling are provided in our GitHub repository (https://github.com/ncats/PANC1)[66] under the MIT license. The repository includes the original license information and copyright statements, ensuring full compliance with the MIT license terms. There are no restrictions on access, and the code is freely available for use and adaptation under the MIT license. The MIT models were based on an adaptation of ComboNet, as described in Jin, W. et al. Deep learning identifies synergistic drug combinations for

treating COVID-19[28]. The original source code for ComboNet is available under the MIT license at https://github.com/wengong-jin/ComboNet. The adapted code retains the original license and copyright statements, with proper attribution included in the scripts.

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

## Acknowledgements
This research was supported in part by the Intramural/Extramural research program of the NCATS, NIH. A.T. and E.N.M. acknowledge the partial support from NIEHS related to the development of mechanism-based approach (grant R41ES033857) and knowledge graph mining (Grant U24ES035214). The authors acknowledge Lu Chen for his assistance with HTS data processing.

## Author contributions
M.P. and S.J. authors contributed equally. M.P. conducted modeling for NCATS, analyzed data, and revised the manuscript. S.J. analyzed and compiled data and drafted the manuscript. E.B. designed assay plates for HTS. W.J. and T.J. conducted modeling for MIT. J.H., T.M., C.M.-F., and A.T. contributed modeling efforts for UNC. Z.I. and K.W. managed compound logistics and performed compound plating. C.K.-T. and S.M. automated and executed the HTS workflow. N.S. provided insights into the biological relevance of the findings. E.N.M. and A.T. supervised the modeling efforts at UNC. R.B. supervised the modeling efforts at MIT. M.F. supervised the PANC-1 assays. A.V.Z. supervised the modeling efforts at NCATS and provided overall study supervision and guidance.

## Funding

## Competing interests
A.T. and E.N.M. are co-founders of Predictive, LLC, which develops novel alternative methodologies and software for toxicity prediction. The remaining authors declare no competing interests.
