## [Transparent Peer Review file · Nature Communications]

AI-driven discovery of synergistic drug combinations against pancreatic cancer

Corresponding Author: Dr Alexey Zakharov

Version 0:

Reviewer comments:

Reviewer #1

(Remarks to the Author)

This study features an in-silico method attempting to provide a strategy for identifying and prioritizing effective drug combinations for pancreatic cancer, a disease resistant to conventional treatments. The authors used Artificial Intelligence (AI) and Machine Learning (ML) to identify efficient synergistic drug combinations, overcoming to the limitation of the experimental methods employing large number of unique chemical combinations. In this study, three independent groups (NCATS, UNC and MIT) deployed distinct AI/ML methodologies based on either structural descriptor-based approach including physical-chemical properties and molecular fingerprints or molecular graph-based approaches, to predict synergistic combinations from a large dataset. Each group trained their models on a set of drug combinations and predicted potential synergies. Indeed, the predictions resulted from different models had minimal overlap and featured high validation rates upon experimental testing using the PANC-1 cancer cell line. The findings underscore the potential value of diverse computational approaches in early virtual screening of potential synergistic candidate drugs. The manuscript is interesting, well presented and I am confident that it will benefit the scientific/medical community working in the field. Still, the overall approach using AI and ML somehow masks the experimental flaws in this study, which exclusively builds on studies in Panc1 cells. This is for me nowadays not state of the art anymore. I can understand the screening nature of the paper and here the panc1 use but the lack of validation and translation to other models prevents publication in its current form. Several concerns are outlined below:

Lane 117: "...modeling approaches were used to nominate 90 compounds, which resulted in a total of 4005 combinations, for further experimental validation". 90 compounds as unique binary combinations would be 4004. Is the extra one coming from positive control with Bortezomib?

Figure 1: the training dataset generation part could be highlighted (e.g. as a box) and labelled to allow for a better discrimination from the actual screening. In addition, it will be clearer for the reader to understand that single agent test was initially used to select promising individual candidates followed by the binary combination screen (496) of selected 32 compounds for combination experiments and both are part of generating training datasets. Furthermore, it is not clear whether Random Forest, Deep Neural network and Graph convolution regression models were employed for all the three AI/ML methods. Here, the authors could mention the three AI/ML methods in the workflow figure for better clarity.

Figure 2: what does "Number" on the Y axis represents in the single-agent dose-response curves? Here, it is not clear how this parameter varies across 32 compounds and different combinations.

Lane 156: "...the original library of 1784 compounds, after omitting the 32 compounds, was used to generate 1,533,876 (all possible) pairwise combinations that serves as the virtual data set to predict prospectively and select new synergistic combination for further biological validation". Here the authors could mention whether the 32 compounds used in the training dataset were omitted later to negate their influence on AI/ML prediction.

Lane 355: It is not clear whether the title "Data generation" is specific for the "Training dataset". Could the authors utilize "Data generation for Training dataset" as a title? Furthermore, they should mention whether the PANC-1 cell line 10x10 assay procedure was common for both training and the experimental validation.

Lane 390: Did the authors include DMSO control and Bortezomib for every 10x10 matrix of the training dataset? Could the authors provide an example plot of training dataset similar to Fig.5 with information on response to negative and positive controls (DMSO and Bortezomib)?

Lane 400: Did the authors use ChemAxon Standardizer to perform structural analysis functions in the both approaches (Descriptor based and Graph-based)?

Lane 411- on synergy calculation: the degree of combinations synergy is referred to as antagonism. However, in lane 137, the non-synergistic is also mentioned as antagonistic. Is this correct?!

To strengthen the findings of the in-silico approach and to delineate the mechanisms of synergy in a biological context, the authors could perform some analyses similar those conducted for the multi-omics investigations (e.g. In-silico Prediction of Synergistic Anti-Cancer Drug Combinations Using Multi-omics Data; Scientific reports, 2019).

In addition, the authors could also discuss about the strategy among different models, particularly given that AI/ML methods do not find overlapping synergistic compounds.

Cytotoxicity is used as a parameter to assess anti-cancer drug performance. In this context, control cell lines (e.g. liver cells) could be employed to test if the observed synergy produced by the drug pairs would cause general cytotoxicity or Panc-1/pancreatic cancer line specific toxicity. Should further biological validation be beyond the scope of this work, the authors could for example discuss how would they further validate their findings in other models than Panc-1 cell line viability assay.

(Remarks on code availability)

Reviewer #2

(Remarks to the Author)

Review Report for "AI-driven discovery of synergistic drug combinations against pancreatic cancer."

Summary

The manuscript investigates the application of AI and machine learning (ML) techniques to identify effective drug combinations against pancreatic cancer, specifically using the PANC-1 cell line. The study involves screening 496 pairwise combinations of 32 anticancer drugs and leveraging different AI/ML methodologies from three independent research groups (NCATS, UNC, MIT) to predict synergistic drug combinations. The results demonstrate significant advances in predicting drug synergy with a notable success rate in experimental validations. Overall, the manuscript presents a comprehensive and innovative approach to drug synergy prediction using AI/ML techniques. The study's findings are significant and relevant to cancer pharmacology. With minor revisions and clarifications, this manuscript will be a valuable contribution to the scientific community.

Major Comments

Comparative Analysis: The study effectively compares different AI/ML approaches. However, it would benefit from a deeper comparative analysis involving additional established methods in drug synergy prediction. This would provide a broader context and could reveal potential advantages or limitations of the current methods.

Methodological Transparency: The manuscript provides a thorough methodological approach and includes GitHub repositories with the code, which enhances transparency and reproducibility. However, more details on the data preprocessing steps and selection criteria for the drug combinations would further support the study's reliability.

Model Performance and Drug Synergism: The overall performance of the models, including the Graph ConvNet, may be impacted by the fact that some drugs have multiple synergistic partners, which may skew the model's performance. The authors could consider evaluating the model upon excluding those drugs to address this. Additionally, methods like stratified sampling or bootstrapping could be employed to ensure robustness in performance metrics.

Random Predictions and Model Validation: The manuscript mentions that the validation strategies used include "compounds out" and "everything out" validation, with y-randomization producing validation statistics consistent with random predictions. This indicates that the authors have considered random prediction baselines. To further strengthen the validation, the authors could include AUC curves for random predictions in the manuscript to visually compare the model's performance against random baselines. This would provide a more explicit demonstration of the model's predictive power and robustness.

Biological Context of Predicted Combinations: Among the predicted combinations, it would be valuable to evaluate further if there is some biological context related to two drugs targeting the same protein or pathway, two divergent pathways, or redundant pathways. This evaluation should be conducted with respect to random drug pairs to determine if the observed synergies are biologically meaningful or due to random chance. Investigating the mechanistic interactions between drugs at the pathway level can provide insights into potential combinatory effects and therapeutic implications.

(Remarks on code availability)

The NCATS PANC1 (<https://github.com/ncats/PANC1>) and Pancreatic Cancer Mixture Modeling (https://github.com/molecularmodelinglab/pancreatic_cancer_mixture_modeling) repositories provide the necessary resources for reproducing the study's results and are highly relevant to the manuscript. In contrast, the ComboNet repository (<https://github.com/wengong-jin/ComboNet>), while well-documented and useful for COVID-19 drug combination studies, is not applicable to pancreatic cancer research as discussed in the manuscript.

Reviewer #3

(Remarks to the Author)

In this manuscript, the authors claim that the traditional process of identifying "synergistic" drug combinations can be time-consuming and laborious. This is because the number of combinations to experiment with can grow exponentially. To address this challenge, they propose an AI-based approach to discover "synergistic" drug combinations from a pool of

1.5M+ potential candidates. The results from three independent groups are used to select a subset of promising drug combinations which are subsequently validated experimentally via cell-based assays.

You will find below some of my comments and suggestions for improving the manuscript.

1. In the introduction, the motivation behind this work is clear at a high level. However, can you be more precise about the challenge you are addressing? Is it that the traditional approach for discovering drug combinations simply does not scale to millions of candidates? Is it that previous hit-rates were too low? Is this a challenge unique to pancreatic cancer? The more precise you can be at the beginning of the manuscript, the better the reader can understand what you are going after.

2. You mention that some work exists in the literature which focuses on using ML to predict the efficacy of drugs against certain types of cancer. However, this comparison is quite superficial. What exactly has previous work achieved? And where do they fall short? And how exactly do you overcome some of their limitations? That is not immediately clear from the introduction. I need some more information to better contextualize the significance of your work.

3. Figures go a long way in helping to communicate your thought-process and methods. While your current Fig. 1 is on the right track, it can be improved to better reflect the flow of the processing of the data. How did you go from A to Z? As this is a critical figure, I would suggest expanding its components (e.g., data source, drug-response matrix, various ML methods, etc.) and providing additional details about each of them.

4. I recommend always providing some motivation behind the choices made throughout the manuscript. For example, you have chosen to classify synergistic drug combinations as those whose gamma value does not exceed 0.95. Is this an established cutoff from the literature? If so, I would make that clear and explain how it was derived. If instead it was based on your own empirical results, then provide some reasoning there. Was it because, for example, it allowed you to split the two classes equally?

5. Since this manuscript has been submitted to Nature Communications, which has a relatively broad audience, I would recommend going the extra mile to define and explain some of the more technical concepts to someone unfamiliar with the domain of cheminformatics. What does synergy mean in this context (e.g., improved response of the drug combination compared to monotherapy)? What are the definitions of the various synergy metrics (e.g., HSA, gamma, etc.)? Why did you opt to use one over the other?

6. While it is great that you had multiple independent groups experiment with different computational methods to identify synergistic drug combinations, I would like to see how previous methods that attempt to predict inhibition scores compare to your proposed approaches. This could be added to Table 1, for example. The reason I ask for this is not to create more work, but rather to help readers appreciate how much better (hopefully) your approach is compared to the status quo.

7. I appreciate the deep-dive into the NSC-AZD drug combination as it allows readers to concretely see the outcome of your entire pipeline. The corresponding discussion around this particular drug combination was also interesting to read. Having said that, I do feel that the Discussion section as it stands is quite scant. What are some of the limitations of your proposed approach(es)? How do they compare to previous work? What do they bring to the table that is different? Why should people even consider adopting this approach moving forward? This might warrant separating your Results section from your Discussion section.

8. As for the explanation of the ML models in the Methods section, I think the readers may benefit from some more implementation details. As a litmus test, ask yourself if you were to reread this paper in a years' time, would you be able to reproduce these experiments in full without frustration? For the NCATS team, what type of sequential model is the DNN? Also, can you please clarify what you mean by having multiple neurons (e.g., 700, 1000, 2000) in the first layer (lines 440 – 442)? Each layer has a fixed number of neurons. Are you referring to the various configurations you experimented with for this particular layer? In a similar vein, did all teams adopt the exact same 32-fold leave-one-compound-out cross validation strategy (with the same seed)?

9. It seems that the UNC model(s) were trained on the binarized gamma labels (using the cutoff of 0.95) whereas the MIT model(s) were trained on binarized inhibition scores. My understanding is that the former represents the synergy of a drug (or drug combination) whereas the latter represents an intermediate score (which can ultimately be used to derive synergy in the case of deriving the gamma metric). If my understanding is correct, can you please clarify why these particular decisions were made and how you went about binarizing the inhibition scores?

10. It is not entirely clear how each of the teams (NCATS, UNC, MIT) went from the trained models to selecting the top 30 drug combinations? I see that UNC has an in-depth description on lines 207 - 224, however this is a bit convoluted and can benefit perhaps from a flow diagram.

For the MIT team, did you simply calculate the synergy score for each of the 1.5M+ combinations, rank them in descending order of synergy, and simply select the top 30 combinations? It would be interesting to see the distribution of synergy scores across the 1.5M+ combinations. Does there happen to be a small cluster of combinations that are more synergistic than the others?

11. The pre-processing steps of any data pipeline can be critical to the success of a project. In the "Compounds standardization" section, you have suggested that you have followed "canonical data curation practices". While these may

be known to researchers in the field, could you please outline these practices in sufficient detail that would enable a non-expert to replicate the study?

12. Each group trained a distinct set of models and applied them to the 1.5M+ drug combinations to ultimately select 30 combinations which they believed were most synergistic. In lines 267 - 274, it appears that you have re-applied these models to the Top 88 drug combinations identified by the previous step. May you explain the motivation behind this evaluation step?

My understanding is that these 88 drug combinations do not have a ground-truth synergy score. And since they were nominated, then you are working under the hypothesis that they might be synergistic. If my understanding is correct, could you help me understand how Table 1 was generated? I would expect to see evaluation metrics like positive predictive value (PPV) and negative predictive value (NPV) but not necessarily a specificity score.

13. I appreciate the two forms of cross-validation that were conducted by the groups (leave-one-compound-out and everything out). The latter is more reflective of how the models would be used in the 'real-world' upon deployment on the 1.5M+ drug combinations, as you are perhaps unlikely to encounter drug compounds that were in your training set. At least for the UNC team, the relatively low performance of the 'everything out' setup begs the question of how confident you are that this particular model will generalize to the 1.5M+ drug combinations?

(Remarks on code availability)

Version 1:

Reviewer comments:

Reviewer #1

(Remarks to the Author)

The authors should be congratulated.

(Remarks on code availability)

Good job

Reviewer #2

(Remarks to the Author)

The authors have satisfactorily addressed all my comments in their revised manuscript, and I am pleased with the current version.

(Remarks on code availability)

The code is provided in Python scripts and Jupyter notebooks, which reproduce the main results of the manuscript.

REVIEWER COMMENTS

Reviewer #1 (Remarks to the Author): Expert in pancreatic cancer therapy and high-throughput drug response screening

This study features an in-silico method attempting to provide a strategy for identifying and prioritizing effective drug combinations for pancreatic cancer, a disease resistant to conventional treatments. The authors used Artificial Intelligence (AI) and Machine Learning (ML) to identify efficient synergistic drug combinations, overcoming to the limitation of the experimental methods employing large number of unique chemical combinations. In this study, three independent groups (NCATS, UNC and MIT) deployed distinct AI/ML methodologies based on either structural descriptor-based approach including physical-chemical properties and molecular fingerprints or molecular graph-based approaches, to predict synergistic combinations from a large dataset. Each group trained their models on a set of drug combinations and predicted potential synergies. Indeed, the predictions resulted from different models a had minimal overlap and featured high validation rates upon experimental testing using the PANC-1 cancer cell line. The findings underscore the potential value of diverse computational approaches in early virtual screening of potential synergistic candidate drugs. The manuscript is interesting, well presented and I am confident that it will benefit the scientific/medical community working in the field. Still, the overall approach using AI and ML somehow masks the experimental flaws in this study, which exclusively builds on studies in Panc1 cells. This is for me nowadays not state of the art anymore. I can understand the screening nature of the paper and here the panc1 use but the lack of validation and translation to other models prevents publication in its current form. Several concerns are outlines below:

Lane 117: "...modeling approaches were used to nominate 90 compounds, which resulted in a total of 4005 combinations, for further experimental validation". 90 compounds as unique binary combinations would be 4004. Is the extra one coming from positive control with Bortezomib?

We apologize for any confusion caused by the wording in the original text and thank the reviewer for pointing it out. To clarify, each of the three groups independently proposed 30 combinations for experimental validation, with two overlaps between the groups. As a result, 88 unique

combinations were tested experimentally (90 total combinations minus the 2 overlaps). The mention of the number 4005 was a mistake in the manuscript and is unrelated to the study. We have corrected the text as follows:

A paragraph in *Introduction* was revised:

“NCATS initiated the study by conducting cell-based assays, screening 1,785 single-agent compounds and identifying the 32 most active ones. NCATS then screened 496 combinations, testing all-vs.-all combinations of the 32 compounds. The three teams used these 496 screening results to train ML models, independently predicting the top 30 synergistic combinations from a virtual library of 1.6 million combinations.”

A paragraph in *Data overview* was revised:

“The study began by selecting 32 compounds from a library of 1,785. Figure 2a illustrates the activity distribution of these compounds in single-agent dose–response curves in PANC-1 cell assays. The IC₅₀ values range between 2 nM and 3 μM, demonstrating variability in activity values. Screening all possible pairwise combinations of 32 compounds, with titration producing 10×10 matrices, generated a modeling dataset of 496 combinations. ”

A paragraph in *Generation of training dataset* was revised:

“Screening the MIPE library in proliferating PANC-1 cells revealed 32 compounds. These compounds, with IC₅₀ values in Fig. 2, show variability and potency, reach efficacy values of at least 50%, and follow curve classes 1.1 and 1.2 [Southall, Noel T., et al. "Enabling the large-scale analysis of quantitative high-throughput screening data." Handbook of Drug Screening. CRC Press, 2016. 456-478]. The 32 compounds produce $32 \times 31 / 2 = 496$ combinations, each tested in 10x10 matrices with nine 1:2 serial dilutions.”

Figure 1: the training dataset generation part could be highlighted (e.g. as a box) and labelled to allow for a better discrimination from the actual screening. In addition, it will be clearer for the

reader to understand that single agent test was initially used to select promising individual candidates followed by the binary combination screen (496) of selected 32 compounds for combination experiments and both are part of generating training datasets. Furthermore, it is not clear whether Random Forest, Deep Neural network and Graph convolution regression models were employed for all the three AI/ML methods. Here, the authors could mention the three AI/ML methods in the workflow figure for better clarity.

We thank the reviewer for this comment and have revised Figure 1 to improve clarity. Specifically, we now include the algorithms in the workflow diagram.

To better distinguish between the generation of training data and the actual screening process, we have added relevant details in both *Introduction* and the *Data overview*, as follows:

The revised section in the *Introduction* reads:

“NCATS initiated the study by conducting cell-based assays, screening 1,785 single-agent compounds and identifying the 32 most active ones. NCATS then screened 496 combinations, testing all-vs.-all combinations of the 32 compounds. The three teams used these 496 screening results to train ML models, independently predicting the top 30 synergistic combinations from a virtual library of 1.6 million combinations. Finally, NCATS tested the synergy of the predicted combinations in cell-based assays, which revealed an average hit rate of 60% across the teams.”

The corresponding part in the *Data overview* reads:

“The study began by selecting 32 compounds from a library of 1,785. Figure 2a illustrates the activity distribution of these compounds in single-agent dose–response curves in PANC-1 cell assays. The IC₅₀ values range between 2 nM and 3 μM, demonstrating variability in activity values. Screening all possible pairwise combinations of 32 compounds, with titration producing 10×10 matrices, generated a modeling dataset of 496 combinations.”

Further down in the *Data overview* subsection:

“The original library of 1,785 compounds produces 1,592,220 combinations. Excluding the 496 combinations from the training set reduces the total to 1,591,724 combinations for the test set. Although the test set omits specific combinations from the training set, it still includes combinations involving compounds from that set. The test dataset includes SMILES representations, IC50 values, and MoAs, but does not include gamma values. Modeling focused on predicting synergy for the test combinations before experimental validation. Three independent research groups conducted separate modeling approaches, each delivering three lists of top 30 synergistic combinations. The selected combinations underwent experimental testing to evaluate the models' accuracy.”

Figure 2: what does “Number” on the Y axis represents in the single-agent dose-response curves? Here, it is not clear how this parameter varies across 32 compounds and different combinations.

We thank the reviewer for this comment. The "Number" on the Y-axis represents the number of compounds corresponding to IC50 values. That is, the total area under the histogram is 32, which is the total number of compounds. We have updated the figure to make this clearer, including a revised Y-axis label: "Number of Compounds." This clarification is also noted in the *Data Overview* subsection:

“Figure 2a illustrates the activity distribution of these compounds in single-agent dose–response curves in PANC-1 cell assays. The IC₅₀ values range between 2 nM and 3 μM, demonstrating variability in activity values.”

To further clarify "how this parameter varies across 32 compounds and different combinations," we re-plotted the histogram using smaller bin widths, which increases the number of bins and provides greater resolution for distinguishing between log(IC₅₀) values. Figure 2a is now updated with this change.

Lane 156: "...the original library of 1784 compounds, after omitting the 32 compounds, was used to generate 1,533,876 (all possible) pairwise combinations that serves as the virtual data set to predict prospectively and select new synergistic combination for further biological validation". Here the authors could mention whether the 32 compounds used in the training dataset were omitted later to negate their influence on AI/ML prediction.

We sincerely thank the reviewer for their careful observation, which prompted us to recheck the data and identify two typographical errors in the reported numbers. We initially stated that the original library contained 1,784 compounds, but the correct number is 1,785. Similarly, the number of combinations in the test set was previously reported as 1,533,876, when the correct figure is 1,591,724. These numbers were corrected throughout the manuscript.

To clarify, the test set was generated from 1,785 compounds, resulting in $1,785 * 1,784 / 2 = 1,592,220$ combinations. After excluding the training set combinations ($32 * 31 / 2 = 496$), the final size of the test set is 1,591,724.

Regarding the reviewer's question about "whether the 32 compounds used in the training dataset were omitted to negate their influence on AI/ML prediction," as outlined in the description above, the test set excludes the training set combinations but still includes compounds from the training set. We have updated *Data overview* to reflect these corrections, as follows:

“The original library of 1,785 compounds produces 1,592,220 combinations. Excluding the 496 combinations from the training set reduces the total to 1,591,724 combinations for the test set. Although the test set omits specific combinations from the training set, it still includes combinations involving compounds from that set. The test dataset includes SMILES representations, IC50 values, and MoAs, but does not include gamma values. Modeling focused on predicting synergy for the test combinations before experimental validation. Three independent research groups conducted separate modeling approaches, each delivering three lists of top 30 synergistic combinations. The selected combinations underwent experimental testing to evaluate the models' accuracy.”

We sincerely apologize for any confusion caused and greatly appreciate the reviewer's attention to detail.

Lane 355: It is not clear whether the title “Data generation” is specific for the “Training dataset”. Could the authors utilize “Data generation for Training dataset” as a title? Furthermore, they should mention whether the PANC-1 cell line 10x10 assay procedure was common for both training and the experimental validation.

We updated the title to *Generation of the training dataset* and clarified that the PANC-1 cell line 10x10 assay procedure was identical for both training and experimental validation. In the subsection *Generation of the training dataset*, we added the following:

"Both the generation of training data and the validation of ML predictions used the PANC-1 cell line 10x10 assay procedure in duplicates."

Additionally, we changed the title of the subsection *Experimental Validation* to *Experimental validation of predictions* and included this passage to emphasize the similarity between the assay procedures in generating training data and validating predictions for the test set:

"Experimental testing of the 88 unique combinations involved duplicated measurements in the PANC-1 cell line assay, using titrations to generate 10×10 dose–response matrices, similar to developing data for ML training."

Lane 390: Did the authors include DMSO control and Bortezomib for every 10x10 matrix of the

training dataset? Could the authors provide an example plot of training dataset similar to Fig.5 with information on response to negative and positive controls (DMSO and Bortezomib)?

Yes, DMSO (negative control) and Bortezomib (positive control) were included in the assay for every 1536-well plate. The experiments were conducted using a 1536-well plate format, with DMSO systematically placed in column 1 and Bortezomib included in columns 2 and 3 of each plate. These controls were consistently applied across all plates used in the high-throughput screening to ensure reproducibility and reliability of the results. We have added an example plot of one plate from the training dataset, similar to Supplementary Fig. 10, showing the response to the negative and positive controls in the supporting information (Supplementary Fig. 11). This image represents a 1536-well screening plate used in the study, illustrating the layout of control wells and screening blocks. DMSO (column 1) shows minimal activity (green), while Bortezomib (columns 2 and 3) exhibits strong inhibition (red). The remaining wells are arranged into 10x10 blocks for compound screening. The color gradient, transitioning from red to black, indicates potent inhibition, with black representing maximal inhibition. Green areas denote little to no inhibitory activity. These sentences were added to *Generation of the training dataset*:

“Each 1536 well plate included a DMSO control (IC_0) and Bortezomib, a well-known cancer drug, as the positive control (IC_{100}). An example plot of the training dataset, including the response to negative and positive controls, is provided in Supplementary Fig. 11. ”

Lane 400: Did the authors use ChemAxon Standardizer to perform structural analysis functions in the both approaches (Descriptor based and Graph-based)?

We thank the reviewer for this comment. Chemical standardization is the first step in obtaining a curated dataset for modeling. We used ChemAxon Standardizer for structural analysis in both descriptor-based and graph-based approaches. We have clarified this in *Compounds standardization*:

“As chemical standardization is the first step in obtaining a curated dataset for modeling, we used ChemAxon Standardizer for structural analysis in both descriptor-based and graph-based approaches.”

Lane 411- on synergy calculation: the degree of combinations synergy is referred to as antagonism. However, in lane 137, the non-synergistic is also mentioned as antagonistic. Is this correct?!

We appreciate the reviewer for highlighting this source of confusion, which arose from a grammatical oversight. Our intent in saying "degree of combination synergy (or antagonism)" was not to equate synergy with antagonism. Instead, we aimed to broaden the discussion to encompass both synergy and antagonism. We now deleted the parenthesis to avoid confusion.

Regarding the terms “synergistic” and “non-synergistic,” it is important to note that the focuses of the ML classification models were to differentiate synergy from two other cases: independence and antagonism. In other words, the ML classification models only predicted whether a combination is synergistic ($\gamma < 0.95$) or not ($\gamma \geq 0.95$), without categorizing the cases where gamma exceeds 0.95. We have emphasized this distinction in two sections of the text to ensure that readers understand the models classify only two cases, not three.

After removing the parentheses in "degree of combination synergy (or antagonism)" and clarifying the differences between synergistic and non-synergistic cases, the paragraph in *Synergy quantification* reads:

“The degree of combination synergy is quantified by comparing the observed compound combination response against the expected response, calculated using a reference model that assumes no interaction between compounds. The commonly-used reference models include the highest single agent (HSA)²⁶, Bliss²⁷, Loewe²⁸ and Zero interaction potency (ZIP) model²⁹. This study analyzed different synergy metrics, such as gamma (γ), beta, and Excess HSA, to assess the reproducibility of duplicated combinations. It chose gamma for its lower sensitivity to variability in duplicates. A previous study determined $\gamma < 0.95$ for synergism through isobolographic assessment of three ibrutinib-drug combinations. Consistently, this study classifies combinations with Gamma scores below 0.95 as synergistic for the ML models, while higher scores indicate non-synergism. The models predicted synergy without differentiating whether non-synergistic combinations were independent or antagonistic.”

Another place where the distinction between synergistic and non-synergistic cases is clarified is in the *Data overview* subsection, as shown below:

“This study focuses on identifying synergistic combinations, disregarding the distinction between additivity and antagonism. Each group developed models using their own methodologies, whether by using gamma scores from the training data or creating bespoke synergy metrics from IC50 values. Nevertheless, the study defines a precise cutoff to evaluate model performance in prospective experiments: Gamma scores below 0.95 indicate synergism, while scores above 0.95 signify non-synergism”

To strengthen the findings of the in-silico approach and to delineate the mechanisms of synergy in a biological context, the authors could perform some analyses similar those conducted for the multi-omics investigations (e.g. In-silico Prediction of Synergistic Anti-Cancer Drug Combinations Using Multi-omics Data; Scientific reports, 2019). In addition, the authors could also discuss about the strategy among different models, particularly given that AI/ML methods do not find overlapping synergistic compounds.

We appreciate the reviewer’s feedback and the reference to the publication by Celebi et al., where they explored synergy mechanisms using experimentally derived and validated biological descriptors, such as drug target information, protein domains, pathways, gene expression profiles, and other molecular features. Our study, however, is limited to descriptors derived from the SMILES representation of compounds, their single-agent activities, and their MoAs. As a result, we lack the extensive data available in the training set of that study. While incorporating this additional information would undoubtedly be valuable, generating it is a significant task and beyond the scope of our work. Nevertheless, to address the reviewer’s comment to the best of our ability, we performed detailed analyses on both the compounds and the MoAs to shed light on the biological aspect of synergy. Specifically, out of the 307 experimentally tested synergistic combinations, we identified the compounds that frequently appear in these pairs. We also determined the most synergistic MoA–MoA combinations in our experiments. These analyses led to the addition of four new figures and an additional table to the manuscript. The corresponding

paragraphs are in the subsection *Exploring the biological relevance of the most synergistic combination in PANC-1*, as follows:

“With the experimental synergy values available, this section shifts its focus from modeling to analyzing the experimental results. Out of 496 training and 88 test combinations, 256 and 51, respectively, have gamma scores below 0.95. These 307 synergistic combinations include several recurring compounds and MoAs. As shown in Fig. 6, Carfilzomib is the most prevalent compound. Notably, combinations involving Carfilzomib exhibit the highest synergy. A similar analysis of MoAs reveals that proteasome inhibition is the most common, followed by PLK1 and HDAC inhibition. HDAC inhibition, despite being less frequent than PLK1 inhibition, results in more favorable Gamma scores.

Previous analyses considered all synergistic combinations with Gamma scores below 0.95 (distribution shown in Fig. 5). Among the 307 combinations, 26 exhibit strong synergy with Gamma scores below 0.5. These span 20 different MoAs, as detailed in Supplementary Table 5. Network analysis (Fig. 6) identifies proteasome–HDAC inhibition as the most frequent synergistic combination, a trend consistent even when considering all synergistic combinations with Gamma scores below 0.95 (Supplementary Fig. 8). Other significant synergistic MoAs in the set of 26 combinations, excluding proteasome inhibition, include HDAC inhibition combined with mutant-p53 activation and HDAC–Survivin inhibition.

Assessing the statistical significance of the emerging MoA–MoA interactions in synergistic combinations remains important. Proteasome–HDAC inhibition occurred 9 times within the 307 synergistic combinations. If this interaction proves the most prevalent, it encourages further exploration to generalize the findings and predict other compounds with these two MoAs. Alternatively, the observed MoA–MoA interactions may have emerged by chance. To test the significance, the process involved comparing these results against random combinations. The analysis took the 64 unique compounds from the 307 synergistic combinations and generated 307 random combinations, repeating this 1,000 times. Proteasome–HDAC inhibition occurred 4.4 times on average, with a standard error of 1.9 (see Supplementary Fig. 9 for a random network map). This comparison shows that the 9 occurrences in the synergistic compounds significantly exceed random expectations. Proteasome–HDAC inhibition gains even more importance because it remains the most frequent interaction with a stricter synergy criterion ($\gamma < 0.5$; see Fig. 6).“

Furthermore, to provide a comparison between our work and that of Celebi et al. and also imply the limitation of our study, we added a paragraph in *Discussion*:

“Comparing the ML results from this study with existing literature provides valuable insights. NCATS observed similar performance among RF, XGBoost, and DNN (see Supplementary Table 2), with a noticeable decline when compound features shifted from Avalon or Morgan fingerprints to RDKit or in-house descriptors. This highlights that, among these three algorithms, feature selection holds greater significance than the algorithm itself. Celebi et al. also reported that XGBoost and RF outperformed linear regression, Lasso, and SVM. Therefore, both studies demonstrate that XGBoost and RF are suitable algorithms for predicting synergy. However, this result contrasts with a study that trained a DNN on a large oncology screen, demonstrating the advantage of deep learning over standard ML models like RFs and SVMs. In general, making direct comparisons between this study and others is challenging due to differences in feature sets and endpoints. For example, the preceding study used chemical and genomic information as input, Celebi et al. used data from the DREAM AstraZeneca-Sanger Drug Combination Prediction Challenge, which incorporates a wider range of information from different cell lines, while the current study used fingerprints from SMILES, MoAs, and single-agent activities in PANC-1 cells. In a similar context, the methods presented in this study may not perform equivalently across different datasets, such as those from other disease areas. Despite the challenge of directly comparing with previous models, this work demonstrates a key strength by rigorously assessing the models' generalizability across 1.6 million combinations, resulting in remarkable hit rates.”

To address the reviewer's comment to “discuss about the strategy among different models”, we clarified the differences between strategies by adding two dedicated paragraphs with more pieces of information in the *Experimental testing of the predictions*, specifically for this comparison:

“Both MIT and UNC used graph convolutional networks, but MIT's superior model performance likely resulted from training on auxiliary data from NCI-ALMANAC, which includes drug synergy data from multiple diseases (4,000 data points). Moreover, UNC incorporated graph convolutional networks only as part of a broader modeling approach and selection process. The

MIT model also stood out for its diverse selection of synergistic combinations. Figure 4 illustrates that the NCATS and UNC models frequently identified the same compounds as synergistic with multiple others. To ensure diversity, MIT deliberately limited the final selection to the top five combinations for each compound, leading to a more varied top 30 list with fewer recurring compounds.

The nomination process, in addition to the algorithms and features, influences the performance of the models. For example, NCATS likely achieved a high PPV by not excluding combinations with recurrent compounds, selecting combinations where a single recurrent compound primarily drives the synergy. In contrast, MIT limited its selection to the top five combinations involving each compound. UNC's three-tier selection process may have contributed to its lower specificity and balanced accuracy.”

Cytotoxicity is used as a parameter to assess anti-cancer drug performance. In this context, control cell lines (e.g. liver cells) could be employed to test if the observed synergy produced by the drug pairs would cause general cytotoxicity or Panc-1/pancreatic cancer line specific toxicity. Should further biological validation be beyond the scope of this work, the authors could for example discuss how would they further validate their findings in other models than Panc-1 cell line viability assay.

We acknowledge the importance of testing for general cytotoxicity. In our study, we tested six synergistic combinations using Human Pancreatic Duct Epithelial (HPDE) cells in both proliferative and confluent assays. As illustrated in the six figures below, all combinations demonstrated doses of the two compounds that were more active in PANC-1 cell lines than in HPDE, suggesting specificity towards cancerous pancreatic cells. While the HPDE results are presented here, a more comprehensive study is in progress and will be detailed in a future publication. In the plots below, the red bars represent the combination response, while the light blue and purple bars indicate the individual compound responses.

Panobinostat+Carfilzomib:

Panobinostat+Daporinad:

Panobinostat+Oprozomib:

Panobinostat+GMX-1778:

Dinaciclib+BI-2536:

Oprozomib+NCGC00188382:

Testing all 307 synergistic combinations, as well as conducting toxicity assessments in cell lines from different tissue types such as liver cells, remains beyond the scope of this study. Nevertheless, we have outlined potential validation strategies across various models in *Discussion*, as follows:

“Testing for general cytotoxicity is crucial for assessing therapeutic compounds and combinations. The synergistic combinations in this study demonstrate efficacy in PANC-1 cell lines, though it

does not include further biological validation. Future research could explore cytotoxicity by using control cell lines and more advanced models. Organoids from pancreatic cancer patients and normal tissue could provide a more accurate representation of the tumor microenvironment, refining evaluations of drug efficacy and specificity. Spheroid cultures, which mimic the three-dimensional structure of tumors, could also help assess compound penetration and activity. In vivo studies with pancreatic cancer mouse models could further evaluate therapeutic potential, observing effects on tumor growth, metastasis, and potential toxicity. These advanced models would guide future research toward clinical trials by providing comprehensive safety and efficacy assessments.”

Reviewer #2 (Remarks to the Author): Expert in cancer bioinformatics, machine learning, drug response prediction, and pancreatic cancer

Review Report for "AI-driven discovery of synergistic drug combinations against pancreatic cancer."

Summary

The manuscript investigates the application of AI and machine learning (ML) techniques to identify effective drug combinations against pancreatic cancer, specifically using the PANC-1 cell line. The study involves screening 496 pairwise combinations of 32 anticancer drugs and leveraging different AI/ML methodologies from three independent research groups (NCATS, UNC, MIT) to predict synergistic drug combinations. The results demonstrate significant advances in predicting drug synergy with a notable success rate in experimental validations. Overall, the manuscript presents a comprehensive and innovative approach to drug synergy prediction using AI/ML techniques. The study's findings are significant and relevant to cancer pharmacology. With minor revisions and clarifications, this manuscript will be a valuable contribution to the scientific community.

Major Comments

Comparative Analysis: The study effectively compares different AI/ML approaches. However, it would benefit from a deeper comparative analysis involving additional established methods in drug synergy prediction. This would provide a broader context and could reveal potential advantages or limitations of the current methods.

We appreciate the reviewer's insightful suggestion. While our primary objective was to identify synergistic drug combinations rather than to conduct a thorough comparison of ML methods, we recognize the value of this comparison. Consequently, we have incorporated a new paragraph in *Discussion*, where we compare our findings with those of previous studies. In this section, we also highlight the inherent challenges in such comparisons due to the multitude of factors influencing ML predictions, including but not limited to the training data. The added paragraph is as follows:

“Comparing the ML results from this study with existing literature provides valuable insights. NCATS observed similar performance among RF, XGBoost, and DNN (see Supplementary Table 2), with a noticeable decline when compound features shifted from Avalon or Morgan fingerprints to RDKit or in-house descriptors. This highlights that, among these three algorithms, feature selection holds greater significance than the algorithm itself. Celebi et al. also reported that XGBoost and RF outperformed linear regression, Lasso, and SVM. Therefore, both studies demonstrate that XGBoost and RF are suitable algorithms for predicting synergy. However, this result contrasts with a study that trained a DNN on a large oncology screen, demonstrating the advantage of deep learning over standard ML models like RFs and SVMs. In general, making direct comparisons between this study and others is challenging due to differences in feature sets and endpoints. For example, the preceding study used chemical and genomic information as input, Celebi et al. used data from the DREAM AstraZeneca-Sanger Drug Combination Prediction Challenge, which incorporates a wider range of information from different cell lines, while the current study used fingerprints from SMILES, MoAs, and single-agent activities in PANC-1 cells. In a similar context, the methods presented in this study may not perform equivalently across different datasets, such as those from other disease areas. Despite the challenge of directly comparing with previous models, this work demonstrates a key strength by rigorously assessing the models' generalizability across 1.6 million combinations, resulting in remarkable hit rates.”

Methodological Transparency: The manuscript provides a thorough methodological approach and includes GitHub repositories with the code, which enhances transparency and reproducibility. However, more details on the data preprocessing steps and selection criteria for the drug combinations would further support the study's reliability.

We appreciate the reviewer's recognition of our transparency. To further enhance it, we are releasing the source code for preparing the training (496 combinations) and test sets (1,591,724 combinations), along with the input data. This includes a CSV file of the screening data for the 32 compounds used to generate the 496 combinations, along with two folders containing synergy data from duplicate experiments. These compounds were selected from 1,785 tested in PANC-1 proliferating cells, with IC₅₀ values shown in Fig. 2, demonstrating both variability and potency. All compounds have efficacy values of at least 50% and complete Hill curves, classified under curve classes 1.1 and 1.2. We revised a paragraph in *Generation of the training dataset*. The paragraph now reads:

“Screening the MIPE library in proliferating PANC-1 cells revealed 32 compounds. These compounds, with IC₅₀ values in Fig. 2, show variability and potency, reach efficacy values of at least 50%, and follow curve classes 1.1 and 1.2. The 32 compounds produce $32 \times 31/2 = 496$ combinations, each tested in 10x10 matrices with nine 1:2 serial dilutions. The concentration range for each compound was selected individually to arrange their IC₅₀ roughly in the middle of that range. Each 1536 well plate included a DMSO control (IC₀) and Bortezomib, a well-known cancer drug, as the positive control (IC₁₀₀). An example plot of the training dataset, including the response to negative and positive controls, is provided in Supplementary Fig.11. Each block was tested in duplicate. Both the generation of training data and the validation of ML predictions used the PANC-1 cell line 10x10 assay procedure in duplicates.”

Model Performance and Drug Synergism: The overall performance of the models, including the Graph ConvNet, may be impacted by the fact that some drugs have multiple synergistic partners, which may skew the model's performance. The authors could consider evaluating the model upon

excluding those drugs to address this. Additionally, methods like stratified sampling or bootstrapping could be employed to ensure robustness in performance metrics.

The reviewer raised an important point. The training data consisted of all-vs.-all combinations of 32 compounds, resulting in each compound being paired with 31 others. Two strategies exist specifically for predicting combinations to address the reviewer's concern: one-compound-out and everything-out. Both NCATS and UNC adopted the one-compound-out strategy, with UNC also utilizing the everything-out strategy. The figure below exemplifies the one-compound-out strategy for 10 combinations derived from 5 compounds. The training data (blue) exclude combinations involving compound C (red), hence the models are less impacted by the fact that drugs have multiple synergistic partners. These strategies, however, limit the application of other sampling approaches, such as stratified and bootstrapping methods. Nevertheless, data imbalance was less of a concern. Out of the 496 training data points, 256 had synergy scores below 0.95, classifying them as synergistic, with 52% synergistic and 48% non-synergistic data, indicating a balanced dataset.

MIT did not use the compounds-out strategies, though it is available as a flag in ComboNet. That is because in a previous study MIT benchmarked that ComboNet generalizes well to novel drug combinations [Jin, Wengong, et al., PNAS, 118 (39), 2021]. Instead, MIT implemented an 80-10-10 train-validation-test split with random splitting, a method proven to deliver reliable outcomes in prior work. This approach directly translated into this study, yielding a significant hit rate (83%) and demonstrating its continued efficacy.

	A	B	C	D	E
A		Train	Test	Train	Train
B			Test	Train	Train
C				Test	Test
D					Train
E					

Random Predictions and Model Validation: The manuscript mentions that the validation strategies used include "compounds out" and "everything out" validation, with y-randomization producing validation statistics consistent with random predictions. This indicates that the authors have considered random prediction baselines. To further strengthen the validation, the authors could include AUC curves for random predictions in the manuscript to visually compare the model's performance against random baselines. This would provide a more explicit demonstration of the model's predictive power and robustness.

We appreciate the reviewer's suggestion to include AUC curves for random predictions to strengthen the validation, as it provides an explicit demonstration of the model's performance against random baselines. As shown in Fig. 3, we have already included a red dashed line in the graphs to indicate random prediction baselines, and this has also been indicated in the figure caption, as below:

“red dashed line, random prediction baseline; ”

Another instance where the results were compared to a random baseline is the newly added MoA analysis, in which synergistic MoAs were evaluated against random combinations. This is presented in a paragraph in the Results section, as follows:

“Assessing the statistical significance of the emerging MoA–MoA interactions in synergistic combinations remains important. Proteasome–HDAC inhibition occurred 9 times within the 307 synergistic combinations. If this interaction proves the most prevalent, it encourages further exploration to generalize the findings and predict other compounds with these two MoAs. Alternatively, the observed MoA–MoA interactions may have emerged by chance. To test the significance, the process involved comparing these results against random combinations. The analysis took the 64 unique compounds from the 307 synergistic combinations and generated 307 random combinations, repeating this 1,000 times. Proteasome–HDAC inhibition occurred 4.4 times on average, with a standard error of 1.9 (see Supplementary Fig. 9 for a random network map). This comparison shows that the 9 occurrences in the synergistic compounds significantly exceed random expectations. Proteasome–HDAC inhibition gains even more importance because it remains the most frequent interaction with a stricter synergy criterion ($\gamma < 0.5$; see Fig. 6).”

Biological Context of Predicted Combinations: Among the predicted combinations, it would be valuable to evaluate further if there is some biological context related to two drugs targeting the same protein or pathway, two divergent pathways, or redundant pathways. This evaluation should be conducted with respect to random drug pairs to determine if the observed synergies are biologically meaningful or due to random chance. Investigating the mechanistic interactions between drugs at the pathway level can provide insights into potential combinatorial effects and therapeutic implications.

To address the reviewer’s comment to the best of our ability, we performed detailed analyses on both the compounds and the MoAs to shed light on the biological aspect of synergy, and compared the results with the case of randomness. Specifically, out of the 307 experimentally tested synergistic combinations, we identified the compounds that frequently appear in these pairs. We also determined the most synergistic MoA–MoA combinations in our experiments and demonstrated that the emerging MoA–MoA holds significance when compared to random combinations. These analyses led to the addition of four new figures and an additional table to the manuscript. The corresponding paragraphs are in the subsection *Exploring the biological relevance of the most synergistic combination in PANC-1*, as follows:

“With the experimental synergy values available, this section shifts its focus from modeling to analyzing the experimental results. Out of 496 training and 88 test combinations, 256 and 51, respectively, have gamma scores below 0.95. These 307 synergistic combinations include several recurring compounds and MoAs. As shown in Fig. 6, Carfilzomib is the most prevalent compound. Notably, combinations involving Carfilzomib exhibit the highest synergy. A similar analysis of MoAs reveals that proteasome inhibition is the most common, followed by PLK1 and HDAC inhibition. HDAC inhibition, despite being less frequent than PLK1 inhibition, results in more favorable Gamma scores.

Previous analyses considered all synergistic combinations with Gamma scores below 0.95 (distribution shown in Fig. 5). Among the 307 combinations, 26 exhibit strong synergy with Gamma scores below 0.5. These span 20 different MoAs, as detailed in Supplementary Table 5. Network analysis (Fig. 6) identifies proteasome–HDAC inhibition as the most frequent synergistic combination, a trend consistent even when considering all synergistic combinations with Gamma scores below 0.95 (Supplementary Fig. 8). Other significant synergistic MoAs in the set of 26 combinations, excluding proteasome inhibition, include HDAC inhibition combined with mutant-p53 activation and HDAC–Survivin inhibition.

Assessing the statistical significance of the emerging MoA–MoA interactions in synergistic combinations remains important. Proteasome–HDAC inhibition occurred 9 times within the 307 synergistic combinations. If this interaction proves the most prevalent, it encourages further exploration to generalize the findings and predict other compounds with these two MoAs. Alternatively, the observed MoA–MoA interactions may have emerged by chance. To test the significance, the process involved comparing these results against random combinations. The analysis took the 64 unique compounds from the 307 synergistic combinations and generated 307 random combinations, repeating this 1,000 times. Proteasome–HDAC inhibition occurred 4.4 times on average, with a standard error of 1.9 (see Supplementary Fig. 9 for a random network map). This comparison shows that the 9 occurrences in the synergistic compounds significantly exceed random expectations. Proteasome–HDAC inhibition gains even more importance because it remains the most frequent interaction with a stricter synergy criterion ($\gamma < 0.5$; see Fig. 6). “

Reviewer #2 (Remarks on code availability):

The NCATS PANC1 (<https://github.com/ncats/PANC1>) and Pancreatic Cancer Mixture Modeling (https://github.com/molecularmodelinglab/pancreatic_cancer_mixture_modeling) repositories provide the necessary resources for reproducing the study's results and are highly relevant to the manuscript. In contrast, the ComboNet repository (<https://github.com/wengong-jin/ComboNet>), while well-documented and useful for COVID-19 drug combination studies, is not applicable to pancreatic cancer research as discussed in the manuscript.

We thank the reviewer for their observations regarding the relevance and applicability of the code repositories. We have updated the GitHub repository (<https://github.com/ncats/PANC1>) for this project with all relevant information, including scripts for data processing and details on ComboNet. The graph convolutional implementation used in this study is an adaptation of ComboNet, originally developed for identifying synergistic drug combinations for treating COVID-19. We have included this variation in the repository, making it relevant to our current project on pancreatic cancer.

Reviewer #3 (Remarks to the Author): Expert in AI in healthcare

In this manuscript, the authors claim that the traditional process of identifying “synergistic” drug combinations can be time-consuming and laborious. This is because the number of combinations to experiment with can grow exponentially. To address this challenge, they propose an AI-based approach to discover “synergistic” drug combinations from a pool of 1.5M+ potential candidates. The results from three independent groups are used to select a subset of promising drug combinations which are subsequently validated experimentally via cell-based assays.

You will find below some of my comments and suggestions for improving the manuscript.

1. In the introduction, the motivation behind this work is clear at a high level. However, can you be more precise about the challenge you are addressing? Is it that the traditional approach for discovering drug combinations simply does not scale to millions of candidates? Is it that previous

hit-rates were too low? Is this a challenge unique to pancreatic cancer? The more precise you can be at the beginning of the manuscript, the better the reader can understand what you are going after.

We thank the reviewer for this insightful comment. We have refined *Introduction* to clearly articulate the specific challenges addressed by our study.

Regarding the scalability issues of traditional methods:

“The process of identifying effective combinations remains challenging due to the sheer number of available and potential drug-like molecules, the quadratic number of possible combinations, and the time-intensive nature of experimental validation. While high-throughput combinatorial screening is an established method for this task, the vast number of unique chemical combinations still limits its efficiency. ”

Regarding whether these challenges are unique to pancreatic cancer:

“This challenge applies not only to pancreatic cancer but to many diseases that require combination therapies. In this context, *in silico* methods present an efficient complement to HTS and *in vitro* screening.”

The challenge cannot be framed as a low hit rate, as previous computational studies did not focus on predicting synergy for PANC-1, leaving no reference point for comparison. Moreover, hit rates cannot be directly compared across studies with differing endpoints. Therefore, although we reference a COVID-19 synergy paper with a 7% hit rate, we did not aim to emphasize outperforming that rate. Here is the sentence where we cite the COVID-19 paper:

“Jin et al. introduced ComboNet, a deep learning architecture that jointly models molecular structure and biological targets. This model predicted 30 combinations, and testing revealed two synergistic combinations in the context of COVID-19 (7% hit rate) ”

2. You mention that some work exists in the literature which focuses on using ML to predict the

efficacy of drugs against certain types of cancer. However, this comparison is quite superficial. What exactly has previous work achieved? And where do they fall short? And how exactly do you overcome some of their limitations? That is not immediately clear from the introduction. I need some more information to better contextualize the significance of your work.

We thank the reviewer for their valuable comment. As mentioned throughout the manuscript, combination therapy can improve the efficacy of treatment through drug synergy and reduce toxicity by requiring lower doses of individual drugs that exhibit synergy. Previous work on using ML for cancer drug discovery has primarily focused on monotherapies and not on drug synergy, making direct comparisons with our work challenging. These studies typically aim to predict the efficacy of individual drugs rather than their potential synergistic effects when combined. There are very few approaches that predict drug synergy and provide retrospective validation, and even fewer that offer prospective validation, particularly across different targets or cell lines. This is a significant gap in the field, as prospective validation is essential for demonstrating the real-world applicability and reliability of predictive models.

Our study is distinct in that it demonstrates the successful implementation of AI/ML techniques to predict drug synergy with prospective validation specifically for pancreatic cancer, an area that remains largely unexplored. This prospective validation is a crucial step forward, as it provides a more reliable assessment of the model's predictive power in a real-world setting. To address the reviewer's comment, we have modified *Introduction* to better highlight these distinctions and the significance of our work. We have also included additional past work that employs computational approaches to predict drug synergy, as follows:

“The DREAM AstraZeneca-Sanger Drug Combination Prediction Challenge provided a comprehensive combinatorial cell line screening dataset, incorporating molecular data such as somatic mutations, copy-number alterations, DNA methylation, and gene expression profiles, as well as compound data including putative drug targets and chemical properties. Jin et al. introduced ComboNet, a deep learning architecture that jointly models molecular structure and biological targets. This model predicted 30 combinations, and testing revealed two synergistic combinations in the context of COVID-19 (7% hit rate). Other studies have introduced

computational methods like DrugComboRanker, DIGRE, and RACS to effectively predict drug pairs for experimental validation. These methods rely on various data types like known disease pathway interactions, post-treatment gene expression, or drug–protein interactions, which can limit their generalizability and predictive accuracy ”

It is important to note that our study focuses on predicting drug synergy, which involves different methodologies and challenges compared to predicting the efficacy of individual drugs.

3. Figures go a long way in helping to communicate your thought-process and methods. While your current Fig. 1 is on the right track, it can be improved to better reflect the flow of the processing of the data. How did you go from A to Z? As this is a critical figure, I would suggest expanding its components (e.g., data source, drug-response matrix, various ML methods, etc.) and providing additional details about each of them.

We thank the reviewer for this comment. We have modified Fig. 1 for more clarity.

4. I recommend always providing some motivation behind the choices made throughout the manuscript. For example, you have chosen to classify synergistic drug combinations as those whose gamma value does not exceed 0.95. Is this an established cutoff from the literature? If so, I would make that clear and explain how it was derived. If instead it was based on your own empirical results, then provide some reasoning there. Was it because, for example, it allowed you to split the two classes equally?

We thank the reviewer for the comment. The choice of using a gamma value of 0.95 as the cutoff for classifying synergistic drug combinations is indeed based on established practices in the literature (reference number 32 in the main manuscript) and is supported by our empirical observations. To address the reviewer’s comment and make it clear for the readers, the paragraph in *Synergy quantification* under *Methods* now has these relevant sentences:

“A previous study determined $\gamma < 0.95$ for synergism through isobolographic assessment of three ibrutinib-drug combinations. Consistently, this study classifies combinations with Gamma scores

below 0.95 as synergistic for the ML models, while higher scores indicate non-synergism. The models predicted synergy without differentiating whether non-synergistic combinations were independent or antagonistic.”

5. Since this manuscript has been submitted to Nature Communications, which has a relatively broad audience, I would recommend going the extra mile to define and explain some of the more technical concepts to someone unfamiliar with the domain of chemoinformatics. What does synergy mean in this context (e.g., improved response of the drug combination compared to monotherapy)? What are the definitions of the various synergy metrics (e.g., HSA, gamma, etc.)? Why did you opt to use one over the other?

We thank the reviewer for their valuable feedback and acknowledge the importance of making technical concepts accessible to a broad audience. In our manuscript, we have aimed to present the information clearly and understandably, while also providing necessary references for more complex terms and concepts for readers who are unfamiliar with the domain and wish to explore them in greater detail. However, due to word limit constraints, it is not possible to fully explain every technical detail within the manuscript. Our approach aligns with the precedent set by previous publications in Nature Communications, where complex topics are introduced with clarity while maintaining brevity, ensuring that interested readers can consult the provided references for a deeper understanding. Nevertheless, to address the reviewer's comment and provide additional clarity for readers, we have included detailed definitions and explanations of key synergy metrics in Supplementary Table 1:

Table S1: Definitions of Key Synergy Metrics in Drug Combination Studies

Metric	Definition
HSA	The Highest Single Agent (HSA) model is a method used to evaluate drug synergy by comparing the effect of a drug combination to the effect of the most effective single agent within that combination. The synergy is assessed by determining whether the combined effect exceeds the maximum effect of the individual drugs when used alone.

Excess HSA	Excess HSA refers to the extent to which the observed effect of a drug combination exceeds the effect predicted by the Highest Single Agent (HSA) model. It quantifies the degree of synergy by measuring how much more effective the combination is compared to the most effective single agent.
Gamma	the Gamma (γ) score is a measure used to quantify the interaction between two or more drugs in a combination. It represents the ratio of the observed effect of the drug combination to the expected effect under a certain model, such as Bliss Independence or Highest Single Agent (HSA). A Gamma score helps to classify the interaction as synergistic, additive, or antagonistic ¹ .
Beta	The Beta (β) score is another metric used to assess drug interactions, particularly in the context of combinatorial drug screening. It often represents the effect size or potency of a drug combination relative to a control or reference effect. Beta scores can be used to compare the strength or efficacy of different drug combinations under various conditions ¹ .

1. Cokol, M. *et al.* Systematic exploration of synergistic drug pairs. *Mol. Syst. Biol.* 7, 544 (2011).

Additionally, *Data overview* includes the following sentences.

“While all metrics confirmed the reproducibility of the assay results and matrix screening technology, the higher correlation of gamma scores led to its selection as the synergy metric. Machine learning used the average gamma scores of each combination for model training. In addition to gamma, the training data includes SMILES representations, IC50 values, and MoAs for individual compounds.”

Finally, we now emphasize that the choice of Gamma was based on its robustness in quantifying synergy scores. This is explicitly outlined in *Synergy quantification*, where it states:

“It chose gamma for its lower sensitivity to variability in duplicates.”

6. While it is great that you had multiple independent groups experiment with different computational methods to identify synergistic drug combinations, I would like to see how previous

methods that attempt to predict inhibition scores compare to your proposed approaches. This could be added to Table 1, for example. The reason I ask for this is not to create more work, but rather to help readers appreciate how much better (hopefully) your approach is compared to the status quo.

We appreciate the reviewer's valuable suggestion to compare our methods with previous approaches. However, previously published models were developed for different cell lines or endpoints, making them unsuitable for direct comparison. To address the reviewer's comments and highlight the challenge of comparison, we have added a paragraph in *Discussion*:

“Comparing the ML results from this study with existing literature provides valuable insights. NCATS observed similar performance among RF, XGBoost, and DNN (see Supplementary Table 2), with a noticeable decline when compound features shifted from Avalon or Morgan fingerprints to RDKit or in-house descriptors. This highlights that, among these three algorithms, feature selection holds greater significance than the algorithm itself. Celebi et al. also reported that XGBoost and RF outperformed linear regression, Lasso, and SVM. Therefore, both studies demonstrate that XGBoost and RF are suitable algorithms for predicting synergy. However, this result contrasts with a study that trained a DNN on a large oncology screen, demonstrating the advantage of deep learning over standard ML models like RFs and SVMs. In general, making direct comparisons between this study and others is challenging due to differences in feature sets and endpoints. For example, the preceding study used chemical and genomic information as input, Celebi et al. used data from the DREAM AstraZeneca-Sanger Drug Combination Prediction Challenge, which incorporates a wider range of information from different cell lines, while the current study used fingerprints from SMILES, MoAs, and single-agent activities in PANC-1 cells. In a similar context, the methods presented in this study may not perform equivalently across different datasets, such as those from other disease areas. Despite the challenge of directly comparing with previous models, this work demonstrates a key strength by rigorously assessing the models' generalizability across 1.6 million combinations, resulting in remarkable hit rates.”

Additionally, we have expanded the introduction to include more detailed citations of previous studies. The updated paragraph now reads:

“Historically, QSAR efforts focused on monotherapies, but recent advancements have shifted towards predicting drug synergies. Cheng et al. introduced a QSAR-based biological network proximity measure to predict drug synergy in hypertension and cancer. Other studies have leveraged machine learning and deep learning techniques for synergy prediction, with Preuer et al. demonstrating the superiority of deep learning over traditional models using a large oncology screen. The DREAM AstraZeneca-Sanger Drug Combination Prediction Challenge provided a comprehensive combinatorial cell line screening dataset, incorporating molecular data such as somatic mutations, copy-number alterations, DNA methylation, and gene expression profiles, as well as compound data including putative drug targets and chemical properties. Jin et al. introduced ComboNet, a deep learning architecture that jointly models molecular structure and biological targets. This model predicted 30 combinations, and testing revealed two synergistic combinations in the context of COVID-19 (7% hit rate). Other studies have introduced computational methods like DrugComboRanker, DIGRE, and RACS to effectively predict drug pairs for experimental validation. These methods rely on various data types like known disease pathway interactions, post-treatment gene expression, or drug–protein interactions, which can limit their generalizability and predictive accuracy.”

7. I appreciate the deep-dive into the NSC-AZD drug combination as it allows readers to concretely see the outcome of your entire pipeline. The corresponding discussion around this particular drug combination was also interesting to read. Having said that, I do feel that the Discussion section as it stands is quite scant. What are some of the limitations of your proposed approach(es)? How do they compare to previous work? What do they bring to the table that is different? Why should people even consider adopting this approach moving forward? This might warrant separating your Results section from your Discussion section.

We appreciate the reviewer’s feedback and recognition of the value in our detailed analysis of the NSC-AZD drug combination. We agree on the importance of a comprehensive discussion, particularly in addressing the limitations of our proposed methods, comparing them with prior work, and highlighting their distinctive contributions.

In response to the reviewer's concerns, we have expanded the relevant sections within *Discussion* to provide a more thorough comparison with previous studies and clarify the unique aspects of our approach. While acknowledging that the methods may not generalize to other disease areas, we emphasize the strength of this study in its prospective validation across 1.6 million combinations. This prospective validation of drug synergy predictions offers a more robust assessment of the models' real-world applicability:

“Comparing the ML results from this study with existing literature provides valuable insights. NCATS observed similar performance among RF, XGBoost, and DNN (see Supplementary Table 2), with a noticeable decline when compound features shifted from Avalon or Morgan fingerprints to RDKit or in-house descriptors. This highlights that, among these three algorithms, feature selection holds greater significance than the algorithm itself. Celebi et al. also reported that XGBoost and RF outperformed linear regression, Lasso, and SVM. Therefore, both studies demonstrate that XGBoost and RF are suitable algorithms for predicting synergy. However, this result contrasts with a study that trained a DNN on a large oncology screen, demonstrating the advantage of deep learning over standard ML models like RFs and SVMs. In general, making direct comparisons between this study and others is challenging due to differences in feature sets and endpoints. For example, the preceding study used chemical and genomic information as input, Celebi et al. used data from the DREAM AstraZeneca-Sanger Drug Combination Prediction Challenge, which incorporates a wider range of information from different cell lines, while the current study used fingerprints from SMILES, MoAs, and single-agent activities in PANC-1 cells. In a similar context, the methods presented in this study may not perform equivalently across different datasets, such as those from other disease areas. Despite the challenge of directly comparing with previous models, this work demonstrates a key strength by rigorously assessing the models' generalizability across 1.6 million combinations, resulting in remarkable hit rates.”

Additionally, we highlighted the novel aspects of our approaches in the context of pancreatic cancer in the beginning of *Discussion*:

“The complexity of pancreatic cancer drove this study to uncover synergistic drug combinations through an integrated approach of computational and experimental methods.”

In accordance with the reviewer's recommendation, we chose to separate the *Results* from the *Discussion* in the structure of the manuscript.

8. As for the explanation of the ML models in the Methods section, I think the readers may benefit from some more implementation details. As a litmus test, ask yourself if you were to reread this paper in a year's time, would you be able to reproduce these experiments in full without frustration? For the NCATS team, what type of sequential model is the DNN? Also, can you please clarify what you mean by having multiple neurons (e.g., 700, 1000, 2000) in the first layer (lines 440 – 442)? Each layer has a fixed number of neurons. Are you referring to the various configurations you experimented with for this particular layer? In a similar vein, did all teams adopt the exact same 32-fold leave-one-compound-out cross validation strategy (with the same seed)?

We thank the reviewer for their comments regarding the reproducibility of our work and the need for more implementation details in *Methods*. To address the reviewer's concerns, we have provided additional scripts and files on GitHub and included more details in the manuscript. With the information now provided, we are confident that readers should be able to reproduce our results without issues.

Regarding the DNN model used by the NCATS team, the "sequential model" refers to a type of neural network model where layers are arranged in a linear order, with each layer's output serving as the input for the next. This is a general term in the field of deep neural networks (DNNs). Recognizing the broad readership of Nature Communications, we have added further references for clarity. Furthermore, the number of neurons was varied across layers: 700, 1000, and 2000 neurons in the first layer, 500 and 700 neurons in the second layer, and 200 and 300 neurons in the third layer. The final model comprised 700, 500, and 300 neurons in the first, second, and third layers, respectively. The corresponding paragraph is now revised with the added information, as follows:

“A nested hyperparameter tuning scheme optimized the hidden layers, testing 700, 1000, and 2000 neurons in the first layer, 500 and 700 in the second, and 200 and 300 in the third. The final DNN architecture used 700, 500, and 300 neurons in the respective layers.”

With respect to the cross-validation strategies, each team employed independent, unbiased approaches tailored to their specific modeling frameworks. While NCATS used a 32-fold one-compound-out cross-validation approach, UNC conducted both one-compound-out and everything out validations. Meanwhile, MIT divided their data into training (80%), validation (10%), and test sets (10%) to develop their optimal model for virtual screening across the 1.6M compound collection. Although the methods varied, each approach was designed independently to maximize the model's predictive power.

9. It seems that the UNC model(s) were trained on the binarized gamma labels (using the cutoff of 0.95) whereas the MIT model(s) were trained on binarized inhibition scores. My understanding is that the former represents the synergy of a drug (or drug combination) whereas the latter represents an intermediate score (which can ultimately be used to derive synergy in the case of deriving the gamma metric). If my understanding is correct, can you please clarify why these particular decisions were made and how you went about binarizing the inhibition scores?

We thank the reviewer for their comment. The observation is correct. The UNC models were trained on binarized gamma labels with $\gamma=0.95$ as the cutoff. The MIT models were trained on a score based on the Bliss combination model, herein referred to as bespoke synergy score. In the subsection *Computational modeling approach 3: MIT*, we refer readers to the original ComboNet paper for additional details:

“Finally, it quantifies the score of combination (A, B) as $c(AB) - s(AB)$, herein referred to as bespoke synergy score, with higher values indicating stronger synergy. Reference ⁴⁴ provides additional details. The score assigns higher values to more synergistic combinations, unlike gamma.”

Regarding ‘why these particular decisions were made’, the three teams were given a list of 496 combinations along with their synergy scores with a gamma threshold of 0.95, IC50 values and MoAs of individual compounds, and their SMILES. Beyond these, each team had full autonomy in utilizing the data for model development and selecting the top 30 combinations for experimental testing. For instance, NCATS incorporated both numerical values of gamma scores and binary labels into their final model, which integrated regression and classification techniques. UNC focused on binary values, while MIT employed ComboNet, which predicts synergy based on the Bliss combination model.

Regarding how we ‘went about binarizing the inhibition scores’, the revised subsection *Synergy quantification* states that:

“A previous study determined $\gamma < 0.95$ for synergism through isobolographic assessment of three ibrutinib-drug combinations. Consistently, this study classifies combinations with Gamma scores below 0.95 as synergistic for the ML models, while higher scores indicate non-synergism. The models predicted synergy without differentiating whether non-synergistic combinations were independent or antagonistic.”

In practice, we accomplished this in Python using the following line of code, where *df* is the Pandas dataframe containing the Gamma scores:

```
df['Label']=(df['Mean Gamma']<0.95).astype(int)
```

10. It is not entirely clear how each of the teams (NCATS, UNC, MIT) went from the trained models to selecting the top 30 drug combinations? I see that UNC has an in-depth description on lines 207 – 224, however this is a bit convoluted and can benefit perhaps from a flow diagram.

We appreciate the reviewer’s suggestion to clarify the selection criteria for the top 30 drug combinations by each team (NCATS, UNC, MIT).

For NCATS, we added these pieces of information to *NCATS subsection* under *Modeling results*:

“The top-performing model used Avalon-2048 fingerprints and combined RF classification with regression, achieving an AUC of 0.78 ± 0.09 (Fig. 3). The model predicted and selected the final top 30 compounds with the highest synergy probabilities.”

MIT's selection was guided by predicted bespoke synergy scores, though they did not directly choose the top 30 to ensure diversity. This clarification has been added to the *MIT* subsection under *Modeling results*:

“The model predicted synergy scores for 1,591,724 combinations and selected the top 30 while ensuring diversity by limiting the final selection to the top five combinations per compound. No compound appeared in more than five combinations.”

It has also been added to *Discussion*:

“To ensure diversity, MIT deliberately limited the final selection to the top five combinations for each compound, leading to a more varied top 30 list with fewer recurring compounds.”

For UNC, we revised the text to improve readability and included an additional table in the main text to clearly display the three tiers of predicted combinations. The revised paragraphs and the table are provided below:

“UNC developed three models using only descriptors and two models combining descriptors with IC50 data. Consensus model predictions were calculated as an average between the three descriptor consensus model predictions, and the two descriptors + IC50 individual model predictions. The top 30 nominated combinations did not come directly from the top 30 ranked lists of either the consensus model or the descriptor-only model. Rather, in addition to model predictions, the selection process followed three supplementary criteria. Table XX lists these three mutually-exclusive selection tiers. The IC50 values of compounds from the test set served as one criterion. Another required that one compound in the pair exists in the training set. The third criterion ensured that the combination corresponds to the most synergistic MoA pairs from the training set.

Using these criteria, the first tier included combinations with consensus scores above 0.7, two active compounds, one compound in the training set, and synergistic MoA pairs, resulting in 12 combinations. The second tier selected the top combinations ranked by descriptor-only predictions, with two active compounds, one compound in the training set, and no synergistic MoAs, adding another 12 combinations. The third tier identified the top six combinations ranked by the consensus model, featuring two active compounds, no compounds from the training set, and synergistic MoA pairs.

Table 1: UNC nomination strategy. Each successive tier nominates combinations not previously nominated by an earlier tier, ensuring the tiers remain mutually exclusive.

Ranks	Count	Combination filters	MoA filter	Ranking models	Ranking criteria
1-12	12	2 active, 1 train	Yes	Consensus	score > 0.7
13-24	12	2 active, 1 train	No	Descriptor-only	12 highest
25-30	6	2 active, 0 train	Yes	Consensus	6 highest

10. For the MIT team, did you simply calculate the synergy score for each of the 1.5M+ combinations, rank them in descending order of synergy, and simply select the top 30 combinations? It would be interesting to see the distribution of synergy scores across the 1.5M+ combinations. Does there happen to be a small cluster of combinations that are more synergistic than the others?

We thank the reviewer for this comment. Additional information has been incorporated to address the reviewer's comments. The selection was guided by predicted bespoke synergy scores, though MIT did not directly choose the top 30. This clarification has been added to the MIT subsection under Modeling results:

“The model predicted synergy scores for 1,591,724 combinations and selected the top 30 while ensuring diversity by limiting the final selection to the top five combinations per compound.”

It has also been added to the Experimental testing of the predictions subsection:

“To ensure diversity, MIT deliberately limited the final selection to the top five combinations for each compound, leading to a more varied top 30 list with fewer recurring compounds.”

Furthermore, we have conducted the requested analysis and included it in the supplementary section. Specifically, Supplementary Fig. 5 shows the distribution of the predicted gamma scores across all 1.6M+ combinations. The distribution is Gaussian and doesn't reveal any clusters of highly synergistic combinations. Furthermore, Supplementary Fig. 6 shows the top 10 most frequent compounds, illustrating their average gamma scores and corresponding frequencies. The data demonstrates that compounds exhibit synergy with a wide range of other compounds. As mentioned earlier, MIT limited the final selection to the top five combinations for each compound.

11. The pre-processing steps of any data pipeline can be critical to the success of a project. In the “Compounds standardization” section, you have suggested that you have followed “canonical data curation practices”. While these may be known to researchers in the field, could you please outline these practices in sufficient detail that would enable a non-expert to replicate the study?

We thank the reviewer for their comment. We acknowledge the importance of clearly outlining the pre-processing steps in any data pipeline. In the *Compounds Standardization* subsection, we have already provided an overview of the key steps involved: salts and solvents were removed from all compounds, followed by the elimination of counterions, large organic compounds ($D_a \geq 2000$), mixtures, and inorganic compounds. Specific chemotypes such as aromatic, nitro groups, sulfo groups, tautomers, and protonation states were standardized using ChemAxon Standardizer software (<https://chemaxon.com/>). Additionally, we have also included a reference that highlights the importance and detailed steps of the data curation process.

Due to word limit constraints, we have aimed to present the most critical aspects of the data curation process while providing references for readers who wish to explore the methods in greater depth. We believe this strikes a balance between providing sufficient information for replication and adhering to the manuscript's word limit.

12. Each group trained a distinct set of models and applied them to the 1.5M+ drug combinations to ultimately select 30 combinations which they believed were most synergistic. In lines 267 – 274, it appears that you have re-applied these models to the Top 88 drug combinations identified by the previous step. May you explain the motivation behind this evaluation step? My understanding is that these 88 drug combinations do not have a ground-truth synergy score. And since they were nominated, then you are working under the hypothesis that they might be synergistic. If my understanding is correct, could you help me understand how Table 1 was generated? I would expect to see evaluation metrics like positive predictive value (PPV) and negative predictive value (NPV) but not necessarily a specificity score.

We appreciate the reviewer's question and would like to clarify that this study involved prospective validation, demonstrating the real-world applicability and reliability of our predictive models. Each group nominated its top 30 combinations, resulting in a total of 88 after de-duplication. These 88 combinations were then experimentally tested. With the experimental synergy (ground truths) now available for these 88 compounds, each group was asked to report their prediction labels for all combinations, including the 30 they initially provided. Therefore, the whole process yielded 88 experimental values (ground truths), 88 predictive values from NCATS, 88 from UNC, and 88 from MIT. Using this data, a table of model metrics was constructed (Table 1). The metrics presented, including sensitivity, specificity, and balanced accuracy, were chosen to offer a comprehensive evaluation of the models' performance. We have now also included PPV and NPV to provide additional insights into the predictive power of the models. Below see Table 1 with PPV and NPV:

Table 2: Performance of three models and a consensus model in predicting synergy of 88 nominated combinations. This analysis follows experimental testing of 88 combinations, with each group reporting their prediction labels for all combinations, including the 30 they initially provided. TP: True Positives, TN: True Negatives, FP: False Positives, FN: False Negatives, Sens: Sensitivity, Spec: Specificity, PPV: Positive Predictive Value, NPV: Negative Predictive Value, AUC: Area Under the Curve.

	TP	TN	FP	FN	Sens	Spec	PPV	NPV	Balanced Accuracy	AUC
NCATS	40	15	22	11	0.78	0.41	0.65	0.58	0.59	0.56
UNC	50	0	37	1	0.98	0.0	0.57	0.00	0.49	0.60
MIT	51	2	35	0	1.0	0.05	0.59	1.00	0.53	0.78
Consensus	51	2	35	0	1.0	0.05	0.59	1.00	0.53	0.67

13. I appreciate the two forms of cross-validation that were conducted by the groups (leave-one-compound-out and everything out). The latter is more reflective of how the models would be used in the “real-world” upon deployment on the 1.5M+ drug combinations, as you are perhaps unlikely to encounter drug compounds that were in your training set. At least for the UNC team, the relatively low performance of the “everything out” setup begs the question of how confident you are that this particular model will generalize to the 1.5M+ drug combinations?

We acknowledge that the everything-out strategy better reflects how the models would be applied in real-world scenarios with over 1.6M drug combinations. However, the 496 combinations used for training are insufficient for this approach. As shown in the schematic figures below on the left, excluding only combinations with Compound C leaves 6 combinations for training and 4 for testing. In contrast, as shown in the right figure, excluding combinations with both C and E reduces the training set to just 3 combinations, with 7 for testing. This highlights the challenges posed by the everything-out strategy and explains why the correct classification rates and positive predictive values were lower in the everything-out scenario compared to the one-compound-out strategy in UNC models. Nevertheless, it also underscores the importance of using multiple validation strategies to better understand the model's strengths and limitations.

	A	B	C	D	E
A		Train	Test	Train	Train
B			Test	Train	Train
C				Test	Test
D					Train
E					

	A	B	C	D	E
A		Train	Test	Train	Test
B			Test	Train	Test
C				Test	Test
D					Test
E					

The one-compound-out and everything-out strategies function solely as cross-validation techniques, without influencing the model's predictions for the 1.6M dataset. For example, if a model produces an AUC value of x using cross-validation 1 and y using cross-validation 2, only one model exists during deployment, regardless of these AUC results. The inclusion of both strategies in this study aimed to demonstrate the comprehensiveness of the modeling process and to highlight the challenge of the everything-out approach. UNC did not rely on cross-validation to choose a subset of models but used all models and averaged the results to generate consensus scores.

Concerns about generalizability remain valid. With a more extensive training dataset—though not available here—the everything-out strategy could be confidently applied across all modeling efforts, allowing for optimization of model parameters, such as features and algorithms. Generalizability is a greater concern when a model is not prospectively tested and is simply presented for deployment. In this study, however, the models were prospectively tested and generalizability was assessed. UNC's models successfully predicted 12 out of 30 combinations, achieving a 40% hit rate.